# POSTERIOR SAMPLING FOR MULTI-AGENT REINFORCEMENT LEARNING: SOLVING EXTENSIVE GAMES WITH IMPERFECT INFORMATION

**Yichi Zhou, Jialian Li, Jun Zhu**[*]
Dept. of Comp. Sci. & Tech., BNRist Center, Institute for AI, Tsinghua University; RealAI
vofhqn@gmail.com,lijialian7@163.com,dcszj@mail.tsinghua.edu.cn

## ABSTRACT

Posterior sampling for reinforcement learning (PSRL) is a useful framework for making decisions in an unknown environment. PSRL maintains a posterior distribution of the environment and then makes planning on an environment sampled from the posterior distribution. Though PSRL works well on single-agent reinforcement learning problems, how to apply PSRL to multi-agent reinforcement learning problems is largely unexplored. In this work, we extend PSRL to two-player zero-sum extensive-games with imperfect information (TEGI), which is a class of multi-agent systems. Technically, we combine PSRL with counterfactual regret minimization (CFR), which is a leading algorithm for TEGI with a known environment. Our main contribution is a novel design of interaction strategies. With our interaction strategies, our algorithm provably converges to the Nash Equilibrium at a rate of $O(\sqrt{\log T/T})$. Empirical results show that our algorithm works well.

## 1 INTRODUCTION

Reinforcement Learning (RL) (Sutton & Barto, 2018) provides a framework for decision-making problems in an unknown environment, such as robotics control. In an RL problem, agents improve their strategies by gaining information from iterative interactions with the environment. One typical target in designing RL algorithms is to reduce the number of interactions needed to find good strategies. Thus, how to reduce the number of samples by designing efficient interaction strategies is one of the key challenges in RL.

Posterior sampling for RL (PSRL) (Strens, 2000) provides a useful framework for deciding how to interact with the environment. PSRL originates from the famous bandit algorithm Thompson Sampling (Russo et al., 2018), which uses samples from posterior distributions of bandit parameters to calculate current policy. PSRL also maintains a posterior distribution for the underlying environment and uses an environment which is sampled from this posterior to compute its interaction strategies. The interaction strategies are then used to interact with the environment to collect data. The design of the interaction strategies depends on the specific property of the task. For example, in a single-agent RL (SARL) problem, PSRL takes the strategy with the maximum expected reward on the sampled environment as the interaction strategy (Osband et al., 2013). Theoretical and empirical results (Osband & Van Roy, 2016) both demonstrate that PSRL is one of the near-optimal methods for SARL. Moreover, although PSRL is a Bayesian-style algorithm, empirical evaluation (Chapelle & Li, 2011) and theoretical analysis on the multi-armed bandit problems (Agrawal & Goyal, 2017) suggest that it also enjoys good performance for a problem with fixed parameters.

However, applying PSRL to multi-agent RL (MARL) tasks requires additional design on the interaction strategies. This is because the goal of MARL is quite different from that of SARL. In an MARL problem, each agent still aims to maximize its own reward, but the reward of an agent's strategy relies not only on the environment, but also on the strategies of other agents. Therefore, in MARL, the goal of learning is generally referred to finding a Nash Equilibrium (NE) where no agent is willing

---

[*]J.Z is the corresponding author.

to deviate its strategy individually. So we should design the interaction strategies with which the agents can find or approximate the NE efficiently.

More specifically, we consider the RL problems in imperfect information extensive games (Osborne & Rubinstein, 1994). Extensive games provide a unified model for sequential decision-making problems in which agents take actions in turn. Imperfect information here means that agents can keep their own private information, such as the private cards in poker games. Games with imperfect information are also fundamental to many practical applications such as economics and security. In particular, we concentrate on two-player zero-sum imperfect information games (TEGI) where there are two players gaining opposite rewards and a chance player to model the transitions of the environment. When the environment (i.e. the transition functions of the chance player and the reward functions) is known, counterfactual regret minimization (CFR) (Zinkevich et al., 2008) is the leading algorithm in approximating the NE in a TEGI. However, in the RL setting where the environment is unknown, CFR is not applicable.

In this work, we present a posterior sampling algorithm for TEGIs with the technique of CFR. That is, we apply CFR to the environment sampled from the posterior distribution. Our main contribution is a novel design of interaction strategies for the RL problem of TEGIs. With the proposed strategies, we show that our algorithm can provably converge to an approximate NE at a rate of $O(\sqrt{\log T/T})$. Empirical results show that our algorithm works well.

## 2 PRELIMINARY

In this section, we formally define the task of two-player zero-sum imperfect information games (TEGI) under Reinforcement Learning; and then briefly review two closely related techniques, namely counterfactual regret minimization (CFR) (Zinkevich et al., 2008) and posterior sampling for reinforcement learning (PSRL) (Osband et al., 2013), which inspire our solution.

### 2.1 PROBLEM FORMULATION OF TEGI

We start from the definition of general $N$-player extensive games (See (Osborne & Rubinstein, 1994, pg. 200) for a formal definition.), which include TEGI as a nontivial special case when $N = 2$.

**Definition 1** (Extensive game). *An extensive game has the following components:*

- *A finite set of players that includes $N$ agents and a chance player $\mathcal{C}$ representing the "Nature" of the game.*

- *A finite set $H$ of sequences that satisfies: 1) The empty sequence is a member of $H$; 2) If a sequence $\{a_1, \cdots, a_k\}$ belongs to $H$, then for all $1 \leq l < k$, $\{a_1, \cdots, a_l\}$ is a member of $H$. Here, each member of $H$ is a history and each component of a history is an action taken by a player. The set of available actions after a history is denoted by $\alpha(h) = \{a : (h, a) \in H\}$ and the set of terminal histories is denoted by $Z$.*

- *A function $P$ such that $P(h)$ is the player who takes the action after history $h$. $H^i$ represents the set of all $h$ that $P(h) = i$.*

- *A function $c^*$ that is the strategy of the chance player, i.e., $c^*(h, a)$ is the probability of action $a$ occurs after $h$ if $P(h) = \mathcal{C}$.*

- *For each player $i$ (besides the chance player), a partition $\mathcal{I}^i$ of $H$: $\mathcal{I}^i$ is an information partition of player $i$; a set $I \in \mathcal{I}^i$ is a subset of $H$ such that if $h_1, h_2 \in I$, then player $i$ cannot distinguish them.*

- *A reward function $r^*$ where $r^*(h, i)$ is the distribution of the reward of player $i$ at $h \in Z$. We assume the rewards are bounded in $[-1, 1]$.*

For convenience, let $A = \max_h |\alpha(h)|$ be the maximal size of actions for one history. A strategy $\sigma^i$ for player $i$ is a mapping from $H^i$ to the distribution over valid actions. We use $\sigma^i(h, a)$ to represent the probability of taking action $a$ at $h \in H^i$ and $\sigma^i(h)$ to be the vector of $\sigma^i(h, a), a \in \alpha(h)$. And a strategy profile $\sigma$ consists of the strategies of all players in $[N] := \{1, \cdots, N\}$, i.e., $\sigma = \{\sigma^i\}_{i \in [N]}$.

We use $\sigma^{-i}$ to refer to the strategies of all players except $i$. Since player $i$ cannot distinguish $h_1, h_2 \in I \in \mathcal{I}^i$, $\sigma^i(h_1)$ and $\sigma^i(h_2)$ must be the same and we denote $\sigma^i(I) = \sigma^i(h_1)$. For the clarity of notation, we abbreviate $(c^*, r^*)$ as $d^*$. Let $u^i(h|\sigma, d^*)$ denote the expected reward of player $i \in [N]$ at history $h$ under strategy $\sigma$. For convenience, let $u^i(\sigma|d^*) = u^i(h_r|\sigma, d^*)$ where $h_r$ is the root of $H$ and $u^i(h|r^*)$ is the expected reward of player $i$ for $h \in Z$.

$\pi_\sigma(h|d^*)$ is the probability of reaching $h$ with $\sigma$ and $c^*$. It is easy to see that we can decompose $\pi_\sigma(h|d^*)$ into the product of the contribution of each player. That is, $\pi_\sigma(h|d^*) = \prod_{i \in [N] \cup \{\mathcal{C}\}} \pi_\sigma^i(h|d^*)$. We use $D(h)$ to refer to the depth of $h$ in the game tree and $D^i(h)$ to refer to the number of $h$'s ancestors whose player is $i$. Obviously, we have $D(h) = 1 + \sum_{i \in \{N\} \cup \{\mathcal{C}\}} D^i(h)$. And let $D = \max_h D(h)$ and $D^i = \max_h D^i(h)$.

With the above notations, then in a TEGI there are two-players besides the chance player (i.e., $N = 2$), player 1 and player 2, and $u^1(h|r^*) + u^2(h|r^*) = 0$ for all histories $h \in Z$.

**Nash Equilibrium and exploitability**: In a multi-agent system, a solution is often referred to a *Nash Equilibrium* (NE) (Osborne & Rubinstein, 1994). In a TEGI, $\sigma = (\sigma^1, \sigma^2)$ is a NE if and only if $u^i(\sigma|d^*) = \max_{\sigma^{*,i}} u^i(\sigma^{*,i}, \sigma^{-i}|d^*)$. Our target is to approximate NE. More specifically, in TEGIs, the approximation error of $\sigma = (\sigma^1, \sigma^2)$ is usually measured by its exploitability:

$$expl(\sigma|d^*) = \max_{\sigma^{*,1}} u^1(\sigma^{*,1}, \sigma^2|d^*) + \max_{\sigma^{*,2}} u^2(\sigma^1, \sigma^{*,2}|d^*). \tag{1}$$

If the environment of a TEGI, i.e. $d^*$, is known for the players, we can directly use counterfactual regret minimization (CFR) (Zinkevich et al., 2008) to minimize the exploitability of this TEGI, as briefly reviewed in Sec. 2.2.

In this paper, we concentrate on the more challenging yet relatively under-explored setting of TEGI, where the environment $d^*$ is unknown and moreover $d^*$ is subject to some intrinsic uncertainty. For example, poker games with unknown deck fall into this setting. In practice, this setting is not uncommon in industrial organisation with unknown entry, exit and firm-specific sources (Ericson & Pakes, 1995). In this setting, players have to interact with the unknown environment to gain sufficient knowledge of the game for making optimal decisions, thereby becoming a reinforcement learning (RL) task. In particular, the task of finding approximate NEs for TEGIs with unknown $d^*$ is a Multi-Agent Reinforcement Learning (MARL) task (Buşoniu et al., 2010).

Moreover, to consider the intrinsic uncertainty of the unknown environment, we adopt a Bayesian formulation for our TEGI task, which can flexibly incorporate the prior information and enjoys various potential benefits (Russo & Van Roy, 2014). Formally, we consider the setting where the chance player and the reward functions follow a prior distribution $\mathbb{P}_0$. That is, the underlying $d^* = (c^*, r^*)$ is sampled from $\mathbb{P}_0(c, r)$. Here $c$ and $r$ are not necessarily independent. After playing $t$ games, players collect some samples from $d^* = (c^*, r^*)$ and they can get the posterior distribution, denoted as $\mathbb{P}_t$. For example, in the case where $r^*(h, i)$ is a Bernoulli distribution and its prior is a Beta distribution, the posterior distribution $\mathbb{P}_t(r)$ is also a Beta distribution. Similarly if the prior for $c^*$ is a Dirichlet distribution, then $\mathbb{P}_t(c)$ is a Dirichlet distribution since $c^*(h)$ is a multinomial distribution for $h \in H^{\mathcal{C}}$.

To solve the above TEGI problems, we present a method that draws inspirations from the solutions for two simplified settings, as briefly reviewed below.

## 2.2 COUNTERFACTUAL REGRET MINIMIZATION (CFR)

As stated above, when the parameters of the game, i.e. $d^* = (c^*, r^*)$, are known[1], counterfactual regret minimization (CFR) (Zinkevich et al., 2008) provides an effective solution to TEGIs with state-of-the-art performance. Formally, CFR is a self-play algorithm that generates a sequence of strategy profiles, $\{\sigma_t\}_{t=1}^T$, by minimizing the following regrets:

$$R_T^{*,i} = \max_{\sigma^i} \sum_{t=1}^T u^i(\sigma^i, \sigma_t^{-i}|d^*) - \sum_{t=1}^T u^i(\sigma_t|d^*).$$

---

[1]In this case, the distribution of $d^*$ degenerates to a single fixed value; therefore no uncertainty on $d^*$.

---

**Algorithm 1** CFR-PSRL

---

$\quad$ **while** $t < T$ **do**
$\qquad$ Sample $d_t$ and $\tilde{d}_t$ from the posterior $\mathbb{P}_t$.
$\qquad$ **for all** $i \in \{1, 2\}$ **do**
$\qquad\quad$ Select $\sigma_t$ by exploiting CFR to minimize the regret: $\max_{\sigma^i} \sum_{t \leq T} u^i(\sigma^i, \sigma_t^{-i} | d_t) - \sum_{t \leq T} u^i(\sigma_t | d_t)$.
$\qquad$ **end for**
$\qquad$ Calculate interaction strategies $\hat{\sigma}_{1,T}$ and $\hat{\sigma}_{2,T}$ with Eq.(6).
$\qquad$ Use $\hat{\sigma}_{1,T}$ and $\hat{\sigma}_{2,T}$ to interact with the environment to gather data and then compute $\mathbb{P}_{t+1}$.
$\quad$ **end while**
$\quad$ Output: $\bar{\sigma} = \frac{1}{t} \sum_{t=1}^{T} \sigma_t$ .

---

For convenience, we write $\bar{\sigma}_T = \frac{1}{T} \sum_{t=1}^{T} \sigma_t$ if $\bar{\sigma}_T^i(I) = \frac{\sum_t \pi_{\sigma_t}^i(I)\sigma_t^i(I)}{\sum_t \pi_{\sigma_t}^i(I)}$. One important observation (Zinkevich et al., 2008) is that in a TEGI the exploitability is:

$$expl(\bar{\sigma}_T | d^*) = \frac{1}{T}\left(R_T^{*,1} + R_T^{*,2}\right). \qquad (2)$$

Therefore, CFR makes $\bar{\sigma}_T$ converge to the NE by minimizing $R_T^{*,1}$ and $R_T^{*,2}$.

### 2.3 POSTERIOR SAMPLING FOR REINFORCEMENT LEARNING (PSRL)

Posterior sampling for reinforcement learning (PSRL) (Osband et al., 2013) provides an effective method for solving RL problems when the environment has uncertainty. Formally, PSRL applies to the Bayesian RL setting with a given prior distribution over the transition and reward function; and the agents can access this prior distribution and then update the posterior distribution using the observations collected from interactions with the environment. PSRL provides a framework on how to select the strategy to interact with the environment under the Bayesian setting. The process of PSRL can be decomposed into two steps: (1) sampling one environment from the posterior and computing strategies for agents according to the sampled environment; (2) using the computed strategies to interact with the underlying environment and updating the posterior distribution with the collected data. The two steps are iterated. The strategies that are used to play games are called interaction strategies.

For different kinds of RL problems, the interaction strategies for PSRL are different. For example, PSRL chooses the strategy with the maximum expected value as the interaction strategy in the single-agent RL problems (SARL). However, this cannot be trivially extended to MARL since the learning goal turns to be the NE. Hence, in order to apply PSRL to our problem of TZEISs, we need to design of proper interaction strategies.

## 3 METHOD

In this section, we formally present our method, which conjoins the merits of PSRL and CFR and can efficiently compute the approximate NE for TEGI tasks. The key is our design of a proper interaction strategy, which can coordinate with CFR to interact with the environment.

Before diving into details, we given an overview of our algorithm. We call two games interacted with the environment as one episode. In episode $t$, we sample a $d_t$ from posterior distribution $\mathbb{P}_t$ and then apply CFR to $d_t$ to get a policy tuple $(\sigma_t^1, \sigma_t^2)$. Then we sample another $\tilde{d}_t$ to calculate the interaction strategies. Then we use the interaction strategies to interact with the environment to collect data and update the posterior. Our algorithm can converge to the NE at a rate of $O(\sqrt{\log(T)/T})$. The time complexity of computing the interaction strategies is linear to $|H|$.

Here we introduce our method in detail. The algorithm is presented in Alg.1. The detailed format of the interaction strategy will be given soon in Eq. (6).

To compute the approximate NE, we adopt a CFR algorithm to minimize the following regret for episode $T$:

$$\hat{R}_T^i = \max_{\sigma^i} \sum_{t \leq T} u^i(\sigma^i, \sigma_t^{-i}|d_t) - \sum_{t \leq T} u^i(\sigma_t|d_t), \tag{3}$$

where $d_t$ is sampled from the posterior distribution at episode $t$ (i.e., $\mathbb{P}_t$). Then we take $\bar{\sigma} = \frac{1}{T} \sum_{t \leq T} \sigma_t$ as the output strategy for our algorithm. Obviously, simply minimizing $\hat{R}_T$ will not make the exploitability $expl(\bar{\sigma}|d^*)$ small, as $d^*$ can be very different with $d_t$, so we need the interaction strategy to be efficient enough to make sure the difference between $d_t$ and $d^*$ is relatively small. The following equation establishes a relation between the exploitability $expl(\bar{\sigma}|d^*)$ and regret $\hat{R}_T^i$:

$$expl(\bar{\sigma}|d^*) = \frac{1}{T} \left( \hat{R}_T^1 + \hat{R}_T^2 + \sum_{i \in \{1,2\}} \sum_{t \leq T} \left( u^i(\sigma_T^{*,i}, \sigma_t^{-i}|d^*) - u^i(\sigma_T'^i, \sigma_t^{-i}|d_t) \right) \right). \tag{4}$$

Here by fixing that the player $-i$ plays the strategy $\sigma_t^{-i}$ at episode $t$,[2] we use $\sigma_T^{*,i} = \arg\max_{\sigma^i} \sum_{t \leq T} u^i(\sigma^i, \sigma_t^{-i}|d^*)$ to denote player $i$'s optimal strategy in the underlying game $d^*$, and $\sigma_T'^i = \arg\max_{\sigma^i} \sum_{t \leq T} u^i(\sigma^i, \sigma_t^{-i}|d_t)$ to denote player $i$'s optimal strategy when the game at episode $t$ is $d_t$. For convenience, let $\mathcal{G}_T^i = \frac{1}{T} \sum_{t \leq T} (u^i(\sigma_T^{*,i}, \sigma_t^{-i}|d^*) - u^i(\sigma_T'^i, \sigma_t^{-i}|d_t))$ denote the gap between exploitability and the regret from CFR. Intuitively, $\sigma_t$ is generated by CFR with a biased knowledge on the environment. The bias can be described by the term $\mathcal{G}_T^i$. As we can minimize $\hat{R}_T^i$ by CFR, we only need to minimize $\mathcal{G}_T^i$ in order to minimize $expl(\bar{\sigma}|d^*)$. Thus, the target of the interaction strategies is to fix the bias, i.e., minimize $\mathcal{G}_T^i$.

The remaining challenge is to design interaction strategies to minimize $\mathcal{G}_T^i$ efficiently. In episode $t$, we first draw $\tilde{d}_t = (\tilde{c}_t, \tilde{r}_t) \sim \mathbb{P}_t$. Then for $i \in \{1, 2\}$, we compute the strategy that maximizes the cumulative reward gaps between games sampled from the posterior:

$$\tilde{\sigma}_t^i = \arg\max_{\sigma^i} \sum_{t'=1}^{t} \left( u^i(\sigma^i, \sigma_{t'}^{-i}|\tilde{d}_t) - u^i(\sigma^i, \sigma_{t'}^{-i}|d_{t'}) \right). \tag{5}$$

**Interaction strategy:** We adopt the following interaction strategies for episode $T$:

$$\hat{\sigma}_{1,T} = (\tilde{\sigma}_T^1, \sigma_T^2) \qquad \text{and} \qquad \hat{\sigma}_{2,T} = (\sigma_T^1, \tilde{\sigma}_T^2) \tag{6}$$

The computation of $\tilde{\sigma}_t^i$ can be implemented in time $O(|H|)$. To make the whole procedure clear, we use a simple toy game to show the game tree at episode $t$ in Fig. 1. We present $d_t$, $\tilde{d}_t$ and $\sigma_t$. The interaction strategy is then calculated from these quantities. It needs to be emphasized that the strategies $\sigma_t^1$ and $\sigma_t^2$ are generated by CFR in episode $t-1$ and they are used as the opponents' strategies in the interaction strategies.

With the interaction strategies $(\tilde{\sigma}_T^1, \sigma_T^2)$ and $(\sigma_T^1, \tilde{\sigma}_T^2)$, we can prove the following bound on the exploitability $expl(\bar{\sigma})$.

**Theorem 1.** *Let $\xi^i = \sum_{j=1}^{D} \sqrt{\max_{\sigma^i} \sum_{I \in \mathcal{I}^i, D(I)=j} \pi_{\sigma^i}^i(I)}$. Use $\mathbb{E}_{d^*}$ to represent the expectation over all the prior distribution $\mathbb{P}_0(d^*)$. If the true game is sampled from a prior $\mathbb{P}_0$ over the chance player nodes and terminal nodes, then for $\bar{\sigma}_T$ computed by Alg. 1, we have*

$$\frac{1}{T}(\hat{R}_T^1 + \hat{R}_T^2) = O\left( \frac{1}{T} \left( (\xi^1 + \xi^2)\sqrt{AT} \right) \right), \tag{7}$$

$$\mathcal{G}_T^i = O\left( \frac{1}{T} \left( \sqrt{|Z|T \ln(|Z|T)} + \sqrt{|H^\mathcal{C}|D^\mathcal{C} AT \ln(|H^\mathcal{C}|T)} \right) \right), \tag{8}$$

$$\mathbb{E}_{d^*} expl(\bar{\sigma}_T|d^*) = O\left( \frac{1}{T} \left( (\xi^1 + \xi^2)\sqrt{AT} + \sqrt{|Z|T \ln(|Z|T)} + \sqrt{|H^\mathcal{C}|D^\mathcal{C} AT \ln(|H^\mathcal{C}|T)} \right) \right). \tag{9}$$

---

[2]Here for two-player games, $-i$ denotes the other player not $i$; for multi-player games, $-i$ generally denotes all players except $i$.

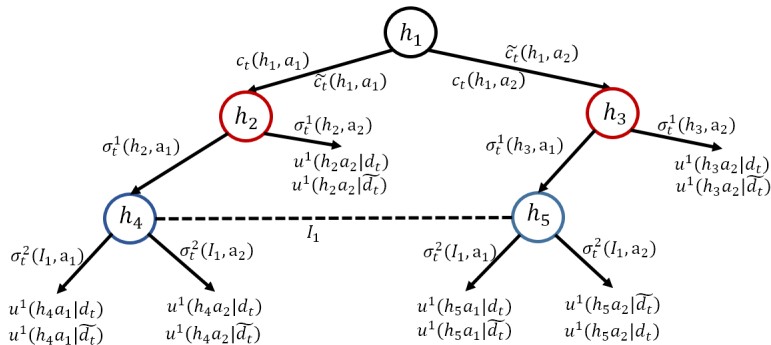

Figure 1: A toy game. Here $P(h_1) = \mathcal{C}$, $h_2, h_3$ are the nodes for player 1 and $h_4, h_5$ are the nodes for player 2. At episode $t$, $d_t$ and $\tilde{d}_t$ are sampled from the posterior distribution, as shown as $c_t, \tilde{c}_t, u^1(\cdot|d_t)$ and $u^1(\cdot|\tilde{d}_t)$. Then $\sigma_t^1$ and $\sigma_t^2$ can be calculated by CFR. Finally we use these parameters on the graph to calculate the interaction strategies with Eq. (5).

Here $\xi^i$ is a game-dependent parameter, related to the structure of the game. Its definition comes from Corrollary 2 in Burch (2018). Under some mild assumptions, we have that $\sqrt{|\mathcal{I}^i|} \leq \xi^i \leq |\mathcal{I}^i|$.

The present theorem is significant at least in the following aspects.

Firstly, the per episode running time is linear to the size of game tree and the bound is sublinear to $T$. Thus, we can expect our algorithm to reach a certain approximate error in a finite time.

Secondly, our theorem holds for any prior distribution over $d^*$. In practical TEGIs, it is possible that the priors for $h_1$ and $h_2$, $h_1, h_2 \in H^\mathcal{C}$, are independent. Our theorem and algorithms can also be applied to such situations.

Lastly, our interaction strategies $\hat{\sigma}_{1,T}$ and $\hat{\sigma}_{2,T}$ only contribute to the bound for $\mathcal{G}_T^i$, which can be treated as the error for interaction strategy's exploring the environment. If we apply PSRL to a single-agent tree game, the Bayesian regret might be considered as some error caused by interacting with the environment. Using the analysis in (Osband et al., 2013), we can get that PSRL enjoys an averaged Bayesian regret bound of order $O(\sqrt{|Z|\ln(|Z|T)/T} + \sqrt{|H^\mathcal{C}|D^\mathcal{C}A\ln(|H^\mathcal{C}|T)/T})$ for a general prior. Therefore, our bound for $\mathcal{G}_T^i$ has a comparable order to the bound for the average Bayesian regret in single-agent tree games.

### 3.1 PROOF SKETCH OF THEOREM 1

Before giving the detailed proof, we introduce some additional notations. For episode $t$, we generate two trajectories by interacting with the environment. More specifically, we use $\mathcal{T}_{i,t}$ ($i \in \{1, 2\}$) to denote the trajectory generated by $\hat{\sigma}_{i,t}$ in environment $d^*$. We use $\mathbb{E}_{\mathcal{T}_{i,t}}$ to denote the expectation over all trajectories for episode $t$. Then we use $\mathcal{T}_{i,t}^\mathcal{C} = \{h_{1,t}^\mathcal{C}, h_{2,t}^\mathcal{C}, ..., h_{m_{i,t},t}^\mathcal{C}\}$ to denote the trajectory for the chance player in episode $t$, and here $m_{i,t}$ denotes the length of $\mathcal{T}_{i,t}^\mathcal{C}$. Furthermore, we denote the terminal node for $\mathcal{T}_{i,t}$ as $z_{i,t}$. Besides, we denote the collection of $\mathcal{T}_{1,1}, \mathcal{T}_{2,1}..., \mathcal{T}_{1,t-1}, \mathcal{T}_{2,t-1}$ and the related rewards as $\mathcal{H}_t$, which represents all the observations before episode $t$. For each history $h$, we further use $n_t(h)$ to denote the count that $h$ has been visited in $\mathcal{H}_t$. Further, we use $\mathbb{E}_{\mathcal{T}_{i,t}}$ to denoting the expectation over all possible trajectories $\mathcal{T}_{i,t}$.

Below we give the key part for the proof. Obviously, we need to bound the regret of CFR, i.e., $\hat{R}_T^i$, and $\mathcal{G}_T^i$. We can directly apply the technique in Theorem 1 of (Burch, 2018, pg. 34) [3] to bound $\hat{R}_T^i$

---

[3]Theorem 1 in (Burch, 2018) was used to analyze regret with the chance player is fixed in different time steps. But it can also be used to bound Eq. (7)

with Eq. (7). Next we show the key part for bounding $\mathcal{G}_T^i$. Using the definition of $\sigma_T'^i$, we have:

$$\mathcal{G}_T^i \leq \frac{1}{T} \sum_{t \leq T} (u^i(\sigma_T^{*,i}, \sigma_t^{-i}|d^*) - u^i(\sigma_T^{*,i}, \sigma_t^{-i}|d_t)))$$

$$\leq \frac{1}{T} \max_{\sigma^i} \sum_{t \leq T} (u^i(\sigma^i, \sigma_t^{-i}|d^*) - u^i(\sigma^i, \sigma_t^{-i}|d_t))).$$

And then, in Lemma 1 and 2, we decompose the above bound into the weighted sum of $|c^*(h) - c_t(h)|$ and $|r^*(h) - r_t(h)|$. And soon later we will show how each term decreases as the number of episodes increases.

**Lemma 1.** *At episode $T$, with $\hat{\sigma}$ defined in Eq. (6), we can upper bound the expectation of $\mathcal{G}_T^i$:*

$$\mathbb{E}_{\mathcal{H}_T} \left\{ \mathbb{E}_{d^*} \left[ \mathcal{G}_T^i \middle| \mathcal{H}_T \right] \right\} \leq \frac{1}{T} \sum_{t=1}^{T} \mathbb{E}_{\mathcal{H}_t} \left\{ \mathbb{E}_{\tilde{d}_t, d^*} \left[ u^i(\tilde{\sigma}_t^i, \sigma_t^{-i}|\tilde{d}_t) - u^i(\tilde{\sigma}_t^i, \sigma_t^{-i}|d^*) \right] \middle| \mathcal{H}_t \right\}$$

$$+ \frac{1}{T} \sum_{t=1}^{T} \mathbb{E}_{\mathcal{H}_t} \left\{ \mathbb{E}_{\tilde{d}_t, d^*} \left[ u^i(\tilde{\sigma}_t^i, \sigma_t^{-i}|d^*) - u^i(\tilde{\sigma}_t^i, \sigma_t^{-i}|d_t) \right] \middle| \mathcal{H}_t \right\}. \quad (10)$$

Lemma 1 decomposes the expectation of $\mathcal{G}_T^i$ into two terms, representing the reward difference between $d^*$ and $\tilde{d}_t$ and the difference between $d^*$ and $d_t$. Below we give an intuitive sketch for bounding the first term $u^i(\tilde{\sigma}_t^i, \sigma_t^{-i}|\tilde{d}_t) - u^i(\tilde{\sigma}_t^i, \sigma_t^{-i}|d^*)$.

**Lemma 2.** *At episode $T$, the expectation of $u^i(\tilde{\sigma}_t^i, \sigma_t^{-i}|\tilde{d}_t) - u^i(\tilde{\sigma}_t^i, \sigma_t^{-i}|d^*)$ can be bounded by the summation of the difference between $d^*$ and $\tilde{d}_t$:*

$$\mathbb{E}_{\mathcal{H}_t} \left\{ \mathbb{E}_{\tilde{d}_t, d^*} \left[ u^i(\tilde{\sigma}_t^i, \sigma_t^{-i}|\tilde{d}_t) - u^i(\tilde{\sigma}_t^i, \sigma_t^{-i}|d^*) \right] \middle| \mathcal{H}_t \right\}$$

$$\leq \mathbb{E}_{\mathcal{H}_t} \left\{ \mathbb{E}_{\tilde{d}_t, d^*} \mathbb{E}_{\mathcal{T}_{i,t}} \left[ \sum_{j=1}^{m_{i,t}} \sum_{a \in \alpha(h)} |\tilde{c}_t(h_{j,t}^{\mathcal{C}}, a) - c^*(h_{j,t}^{\mathcal{C}}, a)| \right] \middle| \mathcal{H}_t \right\}$$

$$+ \mathbb{E}_{\mathcal{H}_t} \left\{ \mathbb{E}_{\tilde{d}_t, d^*} \mathbb{E}_{\mathcal{T}_{i,t}} \left[ u^i(z_{i,t}|\tilde{r}_t) - u^i(z_{i,t}|r^*) \right] \middle| \mathcal{H}_t \right\}. \quad (11)$$

According to the definition of the expectation $\mathbb{E}_{\tilde{d}, d^*} \mathbb{E}_{\mathcal{T}_{i,t}}$, we can see that Eq. (11) is a weighted sum of $|c_t(h) - c^*(h)|$ and $|u^i(h|\tilde{r}_t) - u^i(h|r^*)|$. Recall that $u^i(h|r)$ refers to the expectation of $r(h)$ for player $i$. Intuitively, we can use concentration bound on $|c_t(h) - c^*(h)|$, so for $h$ with a large weight, we should visit it for more times. Notice that the weight in Eq. (11) is essentially the probability of reaching $h$ under our interaction strategy $\hat{\sigma}$ and the real environment $c^*$. Hence if we use $\hat{\sigma}$ to interact with the environment, we can expect our algorithm can visit $h$ with large weight for sufficient times.

To simplify the derivation, we tentatively assume that $\tilde{d}_t$ and $d^*$ are identically distributed for nodes $h_{j,i}^{\mathcal{C}}$ and $z_{i,t}$ conditioning on $\mathcal{H}_t$. That is, for any node $h$, with $Pr$ referring to the probability of some event, we here assume that

$$Pr(d^*|\mathcal{H}_t, h) = Pr(\tilde{d}_t|\mathcal{H}_t, h).$$

In fact this assumption fails when $h$ is reached, because the probability to reach $h$ is influenced by $d^*$ and $\tilde{d}$. We will remove this assumption and provide a rigorous proof in Appendix A. For $(h_{j,i}^{\mathcal{C}}, a)$ and $z_{i,t}$, we can insert the empirical mean estimations $\bar{c}_t(h_{j,i}^{\mathcal{C}}, a)$ and $\bar{u}_t^i(z_{i,t})$ and use the frequentists' concentration bound (Hoeffding, 1994; Weissman et al., 2003). Then for any $\delta \in (0, 1)$, we have the following inequalities:

$$\mathbb{E}_{\mathcal{H}_t} \left\{ \mathbb{E}_{\tilde{d}_t, d^*} \mathbb{E}_{\mathcal{T}_{i,t}} \left[ \sum_{a \in \alpha(h_{j,t}^{\mathcal{C}})} |\tilde{c}(h_{j,t}^{\mathcal{C}}, a) - c^*(h_{j,t}^{\mathcal{C}}, a)| \right] \middle| \mathcal{H}_t \right\} \leq \mathbb{E}_{\mathcal{H}_t} \left[ 2\sqrt{\frac{2\ln(2^A/\delta)}{\max(n_t(h_{j,t}^{\mathcal{C}}), 1)}} \middle| \mathcal{H}_t \right] + 2|H^{\mathcal{C}}|\delta,$$

$$\mathbb{E}_{\mathcal{H}_t} \left\{ \mathbb{E}_{\tilde{d}_t, d^*} \mathbb{E}_{\mathcal{T}_{i,t}} \left[ u^i(h|\tilde{r}_t) - u^i(h|r^*) \right] \middle| \mathcal{H}_t \right\} \leq \mathbb{E}_{\mathcal{H}_t} \left[ 2\sqrt{\frac{2\log(2/\delta)}{\max(n_t(z_{i,t}), 1)}} \middle| \mathcal{H}_t \right] + 4|Z|\delta.$$

Then for a history $h \in Z \cup H^{\mathcal{C}}$, we have $\sum_{n=i}^{n_t(h)} \sqrt{1/i} \leq \sqrt{n_t(h)}$. Using the Jensen's inequality to the summation over $Z$ and $H^{\mathcal{C}}$, we can get the following bound:

$$\sum_{t=1}^{T} \mathbb{E}_{\mathcal{H}_t} \left\{ \mathbb{E}_{\tilde{d}_t, d^*} \left[ u^i(\tilde{\sigma}_t^i, \sigma_t^{-i} | \tilde{d}_t) - u^i(\tilde{\sigma}_t^i, \sigma_t^{-i} | d^*) \right] \Big| \mathcal{H}_t \right\} = O\left( \sqrt{|Z| T \ln(|Z|T)} + \sqrt{|H^{\mathcal{C}}| D^{\mathcal{C}} A T \ln(|H^{\mathcal{C}}|T)} \right).$$

We can apply the same method to $\sum_{t=1}^{T} \mathbb{E}_{H_t} \left\{ \mathbb{E}_{\tilde{d}_t, d^*} \left[ u^i(\tilde{\sigma}_t^i, \sigma_t^{-i} | d^*) - u^i(\tilde{\sigma}_t^i, \sigma_t^{-i} | d_t) \right] \Big| \mathcal{H}_t \right\}$ and get Eq. (8). Combining the results of Eq. (7) and (8), we can finish the proof of the theorem 1.

## 4 RELATED WORK

**Other methods for TEGIs under unknown environment**: There also exist some works on TEGIs under an unknown environment. Fictitious play (FP) (Brown, 1951) is another popular algorithm for approximating NE. In FP, the agent takes the best response to the average strategy of its opponent. Heinrich et al. (2015) extend FP to TEGIs. Though it may be easier to combine FP with other machine learning techniques than CFR, when the chance player is known, the convergence rate of FP is usually worse than CFR variants. Monte Carlo CFR with outcome sampling (MCCFR-OS) (Lanctot et al., 2009) can also be applied to TEGIs to approximate NE in a model-free style. It uses Monte Carlo estimates of the environment to conduct CFR and can converge to the NE. Since it is updated without a model of the environment, it is much less efficient than model-based methods. There is also work that applies SARL methods to TEGIs. For example, Srinivasan et al. (2018) adapts actor-critic to games in a model-free style.

**MDP**: SARL problems are often formalized as the Markov Decision Process (MDP). In the simplest MDP with no transitions, i.e. the Multi-armed bandit problems, the problem-dependent regret upper bound of PSRL (also named Thompson Sampling in bandit problems) has been carefully analyzed (Agrawal & Goyal, 2017). The problem-dependent bounds for general MDP is still an open problem. Besides PSRL, there is another kind of provable algorithms for MDP (Jaksch et al., 2010; Azar et al., 2013) following the Optimism in the Face of Uncertainty principle. They estimate the uncertainty of the underlying MDP and then use the currently optimal policy to interact with the environment.

**Stochastic Games**: the stochastic game (Littman, 1994) is also one kind of multi-agent systems. In a stochastic game, players take actions at each state and then the environment transits to a new state and returns immediate rewards. Nash Q-learning (Hu & Wellman, 2003) converges to approximate NE by extending Q-learning to games, but it lacks finite-time analysis. Some other work (Szepesvári & Littman, 1996; Perolat et al., 2015; Wei et al., 2017) concentrates on two-player zero-sum stochastic games in RL setting. This kind of games don't involve imperfect information, and this makes them different from TEGI.

## 5 EXPERIMENTS

To empirically evaluate our algorithm, we test it on imperfect-information poker games. In this section, we first introduce our baseline methods and then present the details of the games. Finally, we show the results.

We choose three kinds of methods as our baselines. The first one is Fictitious Play (FP) and the second is Monte Carlo CFR with outcome sampling (MCCFR). Both are algorithms for MARL. Thus we can compare the performance of our algorithm and these existing methods. We choose two variants of our algorithm as the other kind of baselines, which is used to compare different choices of interaction strategies. Details of baselines are given below:

- Fictitious self-play (FSP): FSP is another popular algorithm to solve games in the RL setting. In FP, when $d^*$ is known, each player chooses the best response of its opponent's average strategy. When $d^*$ is not known, we need other RL algorithms to learn the best response. We combine FSP with two kinds of RL algorithms: 1) FSP with a fitted-Q iteration algorithm (FSP-fitted-Q): we

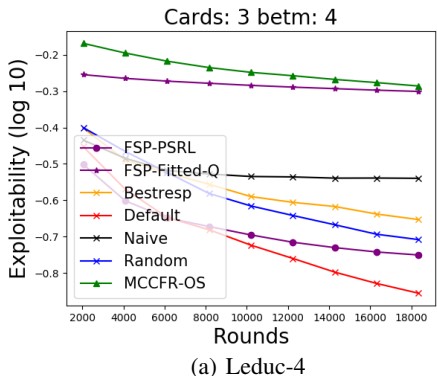 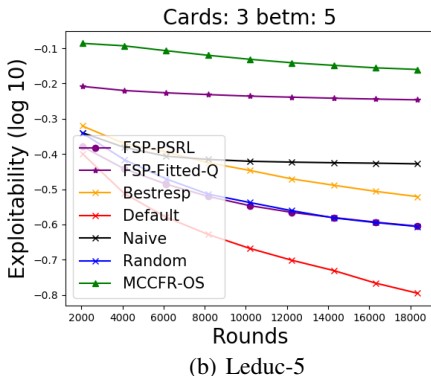

(a) Leduc-4            (b) Leduc-5

Figure 2: Results for different algorithms on variants of Leduc-4 and Leduc-5. Here "default" refers to our algorithm CFR-PSRL.

follow (Heinrich et al., 2015) to use a Fitted-Q iteration to learn the best response. We use the same hyperparameters as reported in Heinrich et al. (2015); 2) FSP with PSRL (FSP-PSRL): we use a combination of FP and PSRL to give a new baseline: In episode $t$, we compute player $i$'s best response under $d_t \sim \mathbb{P}_t$, that is, $\arg\max_{\sigma^i} \sum_{t' < t} u^i(\sigma^i, \sigma_{t'}^{-i} | d_t)$.

- Monte Carlo CFR with outcome sampling (MCCFR-OS): MCCFR-OS uses samples of the game tree to conduct CFR. Estimated counterfactual values are used to update the policies and the average polices can converge to NE. This method can be applied to TEGIs under the RL setting. We use $\epsilon$-greedy with $\epsilon = 0.1$ as the exploration strategy for MCCFR-OS in our experiments.

- Variants of Alg. 1: Though we have proved the convergence of Alg. 1 with interaction strategy $(\tilde{\sigma}^i, \sigma^{-i})$, the proposed method does not necessarily work well in practice. In our experiments, we evaluate four interaction strategies: 1) Random: the players take actions randomly; 2) Naive: the players use the output of the CFR procedure, i.e., $\sigma_t$, to interact with the environment; 3) the best response to $\sigma_t^{-i}$ (Bestresp): $(\tilde{\sigma}_t'^i, \sigma_t^{-i})$ where $\tilde{\sigma}_t'^i$ is the best response to $\sigma_t^{-i}$ under $\tilde{d}_t$ i.e., $\tilde{\sigma}_t'^i = \arg\max_{\sigma^i} u^i(\sigma^i, \sigma_t^{-i} | \tilde{d}_t)$; 4) Default: the interaction strategies in Eq. (6).

We test these algorithms on variants of Leduc Hold'em poker (Southey et al., 2012) which is widely used in imperfect-information game solving. We generate games by keeping the tree structure of the Leduc Hold'em poker and replacing $c$ and $r$ by randomly generated functions. More specifically, when generating the tree structure, to control the sizes of the generated game tree, we restrict each player not to bid more than 4 or 5 times the big blind. The numbers of histories in the generated games are 9435 and 34776 respectively. The reward function $r^i(h)$ is a binary distribution. With a probability $p$ the value of $r^1(h)$ is $-1$ and with probability $1 - p$, the value is 1. The prior $\mathbb{P}_0(r^1(h))$ is a uniform $[0, 1]$ distribution over parameter $p$. Obviously, $r^2(h) = -r^1(h)$. Let $e^d$ denote the vector in $\mathbb{R}^d$ with every element is 1. $c(h)$ is sampled from $Dirichlet(e^{|A(h)|})$.

We generate 20 variants for Leduc(4) and Leduc(5) respectively. And on each generated game, each algorithm updates its strategies for 10000 times, and after each update, it interacts with the environment for 2 rounds of games. The result is shown in Fig. 2. As Fig. 2 shows, the exploitability of naive CFR fails to decrease after 10000 rounds on both Leduc(4) and Leduc(5). This might be caused by the lack of efficient exploration of the environment. MCCFR-OS and FSP-Fitted-Q have poor performances comparing to other algorithms. This may be caused by the data-inefficiency of model-free methods and the inefficient exploration strategies. Random interaction and FP can gradually decrease the exploitability, but our algorithm decrease at a higher speed. Thus the empirical result shows that our algorithm outperforms baselines on the two games.

## 6 CONCLUSIONS AND DISCUSSIONS

In this work, we consider the problem of posterior sampling for TEGIs, which is a class of multi-agent reinforcement learning problems. By a novel design of interaction staregies, we conjoin the

merits of PSRL and CFR and present a provably convergent algorithm for TEGIs. Our algorithm empirically works well.

In the future, there are various directions to improve the result. For example, our bound is a Bayesian bound describing the expected performance. Considering one sample from the prior, Frequentists' methods such as UCBVI (Azar et al., 2013) also give a high probability regret bound for SARL of a similar order to PSRL. Further, comparing with the worst-case bound, the problem-dependent performance is much more important. Though it is possible that our method has a better performance on a specific TEGI than the bound in Theorem 1, our algorithm is very possibly not the best in the sense of problem-dependent performance.

Another direction is that our method heavily relies on the structure of TEGIs and the solution concept of Nash Equilibrium. Thus, further work is needed to extend posterior sampling to more complicated multi-agent systems, such as stochastic games (Littman, 1994) and extensive games with more than two players.

Moreover, the generalization for PSRL is another important but challenging future work direction. It is worth of a systematic investigation to bridge the gap between the provable tabular RL algorithms and PSRL methods with generalization. Bootstrapping might be one possible direction. Osband et al. (2016) applies the principle of PSRL to DQN by using bootstrapping. Another possible direction is to adapt more practical Bayesian inference algorithms to RL tasks.

## ACKNOWLEGEMENT

This work was supported by the National Key Research and Development Program of China (No. 2017YFA0700904), NSFC Projects (Nos. 61620106010, U19B2034, U1811461), Beijing NSF Project (No. L172037), Beijing Academy of Artificial Intelligence (BAAI), Tsinghua-Huawei Joint Research Program, a grant from Tsinghua Institute for Guo Qiang, Tiangong Institute for Intelligent Computing, the JP Morgan Faculty Research Program and the NVIDIA NVAIL Program with GPU/DGX Acceleration.

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

## A  Proof for Theorem 1

Let $\bar{\sigma} = \frac{1}{T}\sum_{t \leq T}\sigma_t$ denote the average strategy tuple up to episode $T$. We can decompose the exploitability at episode $T$ into the CFR regret and an extra exploration term:

$$expl(\bar{\sigma}|d^*) = \frac{1}{T}(\hat{R}_T^1 + \hat{R}_T^2 + \sum_{i \in \{1,2\}}\sum_{t \leq T}(u^i(\sigma_T^{*,i}, \sigma_t^{-i}|d^*) - u^i(\sigma_T'^i, \sigma_t^{-i}|d_t))),$$

where $\hat{R}_T^i$, $\sigma_T^{*,i}$ and $\sigma_T'^i$ are formally defined as:

$$\hat{R}_T^i = \max_{\sigma^i}\sum_{t \leq T}u^i(\sigma^i, \sigma_t^{-i}|d_t) - \sum_{t \leq T}u^i(\sigma_t|d_t),$$

$$\sigma_T^{*,i} = \arg\max_{\sigma^i}\sum_{t \leq T}u^i(\sigma^i, \sigma_t^{-i}|d^*),$$

$$\sigma_T'^i = \arg\max_{\sigma^i}\sum_{t=1}^{T}u^i(\sigma^i, \sigma_t^{-i}|d_t).$$

Here $\hat{R}_T^i$ is the regret from CFR. And $\sigma_T^{*,i}$ and $\sigma_T'^i$ represent the optimal strategies under $d^*$ and $d_t$ respectively, fixing that the its opponent plays the strategy $\sigma_t^{-i}$ at episode $t$.

Recall that $d_t$ is sampled from the posterior distribution $\mathbb{P}_t$. The proof for this decomposition is given below:

*Proof.* Inserting a term $u^i(\sigma_T'^i, \sigma_t^{-i}|d_t)$ for episode $t$, we have:

$$\sum_{t \leq T}u^i(\sigma_T^{*,i}, \sigma_t^{-i}|d^*) = \sum_{t \leq T}(u^i(\sigma_T^{*,i}, \sigma_t^{-i}|d^*) - u^i(\sigma_T'^i, \sigma_t^{-i}|d_t) + u^i(\sigma_T'^i, \sigma_t^{-i}|d_t))$$

$$= \sum_{i,t}(u^i(\sigma_T^{*,i}, \sigma_t^{-i}|d^*) - u^i(\sigma_T'^i, \sigma_t^{-i}|d_t)) + \sum_{t \leq T}u^i(\sigma_t|d_t) + \hat{R}_T^i$$

Moreover, with $u^1(\sigma_t|d_t) + u^2(\sigma_t|d_t) = 0$ and the definition of $expl$, we finish the proof.

$\square$

By directly applying the result in (Burch, 2018), we can upper bound the CFR regret with

$$\hat{R}_T^i \leq \frac{1}{T}\left(\xi^i\sqrt{AT}\right),$$

where $\xi^i = \sum_{j=1}^{D}\sqrt{\max_{\sigma^i}\sum_{I \in \mathcal{I}^i, D(I)=j}\pi_{\sigma^i}^i(I)}$ represents the complexity of the game tree.

For convenience, let $\mathcal{G}_T^i = \frac{1}{T}\sum_{t \leq T}(u^i(\sigma_T^{*,i}, \sigma_t^{-i}|d^*) - u^i(\sigma_T'^i, \sigma_t^{-i}|d_t))$ denote the gap between exploitability and the regret from CFR. Since we have upper bounded $\hat{R}_T^i$, we only need to bound $\mathcal{G}_T^i$ and then we can upper bound the exploitability of the average strategy $\bar{\sigma}$.

Obviously, $\mathcal{G}_T^i$ represents the difference between $d^*$ and $d_t$. Thus we need to design a suitable interaction strategy to make sure that $\mathcal{G}_T^i$ is small. Using the definition of $\sigma_T'^i$, we have

$$\mathcal{G}_T^i \leq \frac{1}{T}\sum_{t \leq T}(u^i(\sigma_T^{*,i}, \sigma_t^{-i}|d^*) - u^i(\sigma_T^{*,i}, \sigma_t^{-i}|d_t)))$$

$$\leq \frac{1}{T}\max_{\sigma^i}\sum_{t \leq T}(u^i(\sigma^i, \sigma_t^{-i}|d^*) - u^i(\sigma^i, \sigma_t^{-i}|d_t))).$$

We select the interaction strategy as follows. First, we draw $\tilde{d}_t \sim \mathbb{P}_t$. For $i \in \{1, 2\}$, we compute

$$\tilde{\sigma}_t^i = \arg\max_{\sigma^i} \sum_{t' \le t} u^i(\sigma^i, \sigma_{t'}^{-i} | \tilde{d}_t) - u^i(\sigma^i, \sigma_{t'}^{-i} | d_{t'}). \tag{12}$$

And then we use $(\tilde{\sigma}_t^1, \sigma_t^2)$ and $(\tilde{\sigma}_t^2, \sigma_t^1)$ to interact with the environment. Following lemma provides an upper bound on $\mathcal{G}_t^i$. Then we get below lemma:

**Lemma 3.** *With $\tilde{\sigma}_t^i$ defined in Eq. (12), the expectation of $\mathcal{G}_T^i$ can be bounded by:*

$$\mathbb{E}_{H_T} \left\{ \mathbb{E}_{d^*, d_T} \left[ \mathcal{G}_T^i \Big| H_T \right] \right\} \le \frac{1}{T} \sum_{t=1}^{T} \mathbb{E}_{H_t} \left\{ \mathbb{E}_{\tilde{d}_t, d^*} \left[ u^i(\tilde{\sigma}_t^i, \sigma_t^{-i} | \tilde{d}_t) - u^i(\tilde{\sigma}_t^i, \sigma_t^{-i} | d^*) \right] \Big| \mathcal{H}_t \right\}$$

$$+ \frac{1}{T} \sum_{t=1}^{T} \mathbb{E}_{H_t} \left\{ \mathbb{E}_{\tilde{d}_t, d^*, d_t} \left[ u^i(\tilde{\sigma}_t^i, \sigma_t^{-i} | d^*) - u^i(\tilde{\sigma}_t^i, \sigma_t^{-i} | d_t) \right] \Big| \mathcal{H}_t \right\}. \tag{13}$$

This lemma decompose $\mathcal{G}_T^i$ into two terms, which can be bounded with careful analysis of the posterior distribution. We first give the proof for this lemma.

*Proof.* We have

$$\mathbb{E}_{\mathcal{H}_T} \left\{ \mathbb{E}_{d^*, d_T} \left[ \max_{\hat{\sigma}^i} \left( \sum_{t=1}^{T} (u^i(\hat{\sigma}^i, \sigma_t^{-i} | d^*) - u^i(\hat{\sigma}^i, \sigma_t^{-i} | d_t)) \right) \right] \Big| \mathcal{H}_T \right\}$$

$$\overset{①}{=} \mathbb{E}_{H_T} \left\{ \mathbb{E}_{\tilde{d}_T, d_T} \left[ \max_{\tilde{\sigma}^i} \left( \sum_{t=1}^{T} (u^i(\tilde{\sigma}^i, \sigma_t^{-i} | \tilde{d}_T) - u^i(\tilde{\sigma}^i, \sigma_t^{-i} | d_t)) \right) \right] \Big| \mathcal{H}_T \right\}$$

$$\overset{②}{=} \mathbb{E}_{\mathcal{H}_T} \left\{ \mathbb{E}_{\tilde{d}_T, d_T} \left[ \sum_{t=1}^{T} (u^i(\tilde{\sigma}_T^i, \sigma_t^{-i} | \tilde{d}_T) - u^i(\tilde{\sigma}_T^i, \sigma_t^{-i} | d_t)) \right] \Big| \mathcal{H}_T \right\}$$

$$= \mathbb{E}_{\mathcal{H}_T} \left\{ \mathbb{E}_{\tilde{d}_T} \left[ \sum_{t=1}^{T-1} (u^i(\tilde{\sigma}_T^i, \sigma_t^{-i} | \tilde{d}_T) - u^i(\tilde{\sigma}_T^i, \sigma_t^{-i} | d_t)) \right] \Big| \mathcal{H}_T \right\}$$

$$+ \mathbb{E}_{\mathcal{H}_T} \left\{ \mathbb{E}_{\tilde{d}_T, d_T} \left[ u^i(\tilde{\sigma}_T^i, \sigma_T^{-i} | \tilde{d}_T) - u^i(\tilde{\sigma}_T^i, \sigma_T^{-i} | d_T) \right] \Big| \mathcal{H}_T \right\}$$

$$\overset{③}{\le} \mathbb{E}_{\mathcal{H}_T} \left\{ \mathbb{E}_{\tilde{d}_T} \left[ \max_{\tilde{\sigma}^i} \left( \sum_{t=1}^{T-1} (u^i(\tilde{\sigma}^i, \sigma_t^{-i} | \tilde{d}_T) - u^i(\tilde{\sigma}^i, \sigma_t^{-i} | d_t)) \right) \right] \Big| \mathcal{H}_T \right\}$$

$$+ \mathbb{E}_{H_T} \left\{ \mathbb{E}_{\tilde{d}_T, d_T} \left[ u^i(\tilde{\sigma}_T^i, \sigma_T^{-i} | \tilde{d}_T) - u^i(\tilde{\sigma}_T^i, \sigma_T^{-i} | d_T) \right] \Big| \mathcal{H}_T \right\}$$

$$\overset{④}{=} \mathbb{E}_{\mathcal{H}_{T-1}} \left\{ \mathbb{E}_{\tilde{d}_{T-1}, d_{T-1}} \left[ \sum_{t=1}^{T-1} (u^i(\tilde{\sigma}_{T-1}^i, \sigma_t^{-i} | \tilde{d}_{T-1}) - u^i(\tilde{\sigma}_{T-1}^i, \sigma_t^{-i} | d_t)) \right] \Big| \mathcal{H}_{T-1} \right\}$$

$$+ \mathbb{E}_{\mathcal{H}_T} \left\{ \mathbb{E}_{\tilde{d}_T, d_T} \left[ u^i(\tilde{\sigma}_T^i, \sigma_T^{-i} | \tilde{d}_T) - u^i(\tilde{\sigma}_T^i, \sigma_T^{-i} | d_T) \right] \Big| \mathcal{H}_T \right\}$$

$$\overset{⑤}{\le} \sum_{t=1}^{T} \mathbb{E}_{\mathcal{H}_t} \left\{ \mathbb{E}_{\tilde{d}_t, d_t} \left[ u^i(\tilde{\sigma}_t^i, \sigma_t^{-i} | \tilde{d}_t) - u^i(\tilde{\sigma}_t^i, \sigma_t^{-i} | d_t) \right] \Big| \mathcal{H}_t \right\}$$

$$\overset{⑥}{=} \sum_{t=1}^{T} \mathbb{E}_{\mathcal{H}_t} \left\{ \mathbb{E}_{\tilde{d}_t, d^*} \left[ u^i(\tilde{\sigma}_t^i, \sigma_t^{-i} | \tilde{d}_t) - u^i(\tilde{\sigma}_t^i, \sigma_t^{-i} | d^*) \right] \Big| \mathcal{H}_t \right\}$$

$$+ \sum_{t=1}^{T} \mathbb{E}_{\mathcal{H}_t} \left\{ \mathbb{E}_{\tilde{d}_t, d^*, d_t} \left[ u^i(\tilde{\sigma}_t^i, \sigma_t^{-i} | d^*) - u^i(\tilde{\sigma}_t^i, \sigma_t^{-i} | d_t) \right] \Big| \mathcal{H}_t \right\}.$$

Here equality ① holds since $d^*$ and $\tilde{d}_T$ are identical distributed conditioning on $\mathcal{H}_t$. Equation ② holds due to the definition of $\sigma_T^{\tilde{i}}$. Then we get ③ with the maximum function. Equality ④ holds

with the definition of $\tilde{\sigma}_{T-1}^i$. Inequality ⑤ holds by using the induction to episode $t$. Finally we get equality ⑥ by inserting $u^i(\tilde{\sigma}_t^i, \sigma_t^{-i}|d^*)$.

Therefore, we finish the proof. □

Then we only to give upper bounds for the above two terms. Also mentioned in sec.3.1, we introduce some additional notations. For episode $t$, we generate two trajectories by interacting with the environment. More specifically, we use $\mathcal{T}_{i,t}$ ($i \in \{1, 2\}$) to denote the trajectory generated by $\hat{\sigma}_{i,t}$ with $d^*$. We use $\mathbb{E}_{\mathcal{T}_{i,t}}$ to denote the expectation over all trajectories for episode $t$. Then we denote $\mathcal{T}_{i,t}^{\mathcal{C}} = \{h_{1,t}^{\mathcal{C}}, h_{2,t}^{\mathcal{C}}, ..., h_{m_{i,t},t}^{\mathcal{C}}\}$ the trajectory for the chance player in episode $t$, and here $m_{i,t}$ denotes the length of $\mathcal{T}_{i,t}^{\mathcal{C}}$. Furthermore, we denote the terminal node for episode $t$ as $z_{i,t}$. Besides, we denote the collection of $\mathcal{T}_{1,1}, \mathcal{T}_{2,1}..., \mathcal{T}_{1,t-1}, \mathcal{T}_{2,t-1}$ as $\mathcal{H}_t$, which represents all the observations before episode $t$. For each history $h$, we further use $n_t(h)$ to denote the count that $h$ has been visited in $\mathcal{H}_t$. Further, we use $\mathbb{E}_{\mathcal{T}_{i,t}}$ to denoting the expectation over all possible trajectories $\mathcal{T}_{i,t}$.

Then we concentrate on the first term $\sum_{t=1}^T \mathbb{E}_{\mathcal{H}_t} \left\{ \mathbb{E}_{\tilde{d}_t, d^*} \left[ u^i(\tilde{\sigma}_t^i, \sigma_t^{-i}|\tilde{d}_t) - u^i(\tilde{\sigma}_t^i, \sigma_t^{-i}|d^*) \right] \Big| \mathcal{H}_t \right\}$ and the second term has similar proof. Since the strategy tuple is the same for the two utilities, we can decompose their difference with below lemma.

**Lemma 4.** *At episode $t$, the expectation of $u^i(\tilde{\sigma}_t^i, \sigma_t^{-i}|\tilde{d}_t) - u^i(\tilde{\sigma}_t^i, \sigma_t^{-i}|d^*)$ can be upper bounded:*

$$\mathbb{E}_{\mathcal{H}_t} \left\{ \mathbb{E}_{\tilde{d}_t, d^*} \left[ u^i(\tilde{\sigma}_t^i, \sigma_t^{-i}|\tilde{d}_t) - u^i(\tilde{\sigma}_t^i, \sigma_t^{-i}|d^*) \right] \Big| \mathcal{H}_t \right\}$$

$$= \mathbb{E}_{\mathcal{H}_t} \left\{ \mathbb{E}_{\tilde{d}_t, d^*} \mathbb{E}_{\mathcal{T}_{i,t}} \left[ \sum_{j=1}^{m_{i,t}} \sum_{a \in \alpha(h)} (\tilde{c}_t(h_{j,t}^{\mathcal{C}}, a) - c^*(h_{j,t}^{\mathcal{C}}, a)) u^i(h_{j,t}^{\mathcal{C}}|\tilde{\sigma}_t^i, \sigma_t^{-i}, \tilde{d}_t) \right] \Big| \mathcal{H}_t \right\}$$

$$+ \mathbb{E}_{\mathcal{H}_t} \left\{ \mathbb{E}_{\tilde{d}_t, d^*} \mathbb{E}_{\mathcal{T}_{i,t}} \left[ u^i(z_{i,t}|\tilde{r}_t) - u^i(z_{i,t}|r^*) \right] \Big| \mathcal{H}_t \right\}.$$

*Proof.* From the root node to $h_{1,t}^{\mathcal{C}}$, players take actions according to $(\tilde{\sigma}_t^i, \sigma_t^{-i})$. Thus we should have

$$\mathbb{E}_{\mathcal{H}_t} \left\{ \mathbb{E}_{\tilde{d}_t, d^*} \left[ u^i(\tilde{\sigma}_t^i, \sigma_t^{-i}|\tilde{d}_t) - u^i(\tilde{\sigma}_t^i, \sigma_t^{-i}|d^*) \right] \Big| \mathcal{H}_t \right\}$$

$$= \mathbb{E}_{\mathcal{H}_t} \left\{ \mathbb{E}_{\tilde{d}_t, d^*} \mathbb{E}_{\mathcal{T}_{i,t}} \left[ u^i(h_{1,t}^{\mathcal{C}}|\tilde{\sigma}_t^i, \sigma_t^{-i}, \tilde{d}_t) - u^i(h_{1,t}^{\mathcal{C}}|\tilde{\sigma}_t^i, \sigma_t^{-i}, d^*) \right] \Big| \mathcal{H}_t \right\}$$

$$= \mathbb{E}_{\mathcal{H}_t} \left\{ \mathbb{E}_{\tilde{d}_t, d^*} \mathbb{E}_{\mathcal{T}_{i,t}} \left[ \sum_{a \in \alpha(h_{1,t}^{\mathcal{C}})} (\tilde{c}_t(h_{1,t}^{\mathcal{C}}, a) u^i(h_{1,t}^{\mathcal{C}}|\tilde{\sigma}_t^i, \sigma_t^{-i}, \tilde{d}_t) - c^*(h_{1,t}^{\mathcal{C}}, a) u^i(h_{1,t}^{\mathcal{C}}|\tilde{\sigma}_t^i, \sigma_t^{-i}, d^*)) \right] \Big| \mathcal{H}_t \right\}$$

$$\overset{①}{=} \mathbb{E}_{\mathcal{H}_t} \left\{ \mathbb{E}_{\tilde{d}_t, d^*} \mathbb{E}_{\mathcal{T}_{i,t}} \left[ \sum_{a \in \alpha(h_{1,t}^{\mathcal{C}})} (\tilde{c}_t(h_{1,t}^{\mathcal{C}}, a) - c^*(h_{1,t}^{\mathcal{C}}, a)) u^i(h_{1,t}^{\mathcal{C}}|\tilde{\sigma}_t^i, \sigma_t^{-i}, \tilde{d}_t) \right] \Big| \mathcal{H}_t \right\}$$

$$+ \mathbb{E}_{\mathcal{H}_t} \left\{ \mathbb{E}_{\tilde{d}_t, d^*} \mathbb{E}_{\mathcal{T}_{i,t}} \left[ \sum_{a \in \alpha(h_{1,t}^{\mathcal{C}})} c^*(h_{1,t}^{\mathcal{C}}, a) (u^i(h_{1,t}^{\mathcal{C}}|\tilde{\sigma}_t^i, \sigma_t^{-i}, \tilde{d}_t) - u^i(h_{1,t}^{\mathcal{C}}|\tilde{\sigma}_t^i, \sigma_t^{-i}, d^*)) \right] \Big| \mathcal{H}_t \right\}$$

$$\overset{②}{=} \mathbb{E}_{\mathcal{H}_t} \left\{ \mathbb{E}_{\tilde{d}_t, d^*} \mathbb{E}_{\mathcal{T}_{i,t}} \left[ \sum_{a \in \alpha(h_{1,t}^{\mathcal{C}})} (\tilde{c}_t(h_{1,t}^{\mathcal{C}}, a) - c^*(h_{1,t}^{\mathcal{C}}, a)) u^i(h_{1,t}^{\mathcal{C}}|\tilde{\sigma}_t^i, \sigma_t^{-i}, \tilde{d}_t) \right] \Big| \mathcal{H}_t \right\}$$

$$+ \mathbb{E}_{\mathcal{H}_t} \left\{ \mathbb{E}_{\tilde{d}_t, d^*} \mathbb{E}_{\mathcal{T}_{i,t}} \left[ u^i(h_{2,t}^{\mathcal{C}}|\tilde{\sigma}_t^i, \sigma_t^{-i}, \tilde{d}_t) - u^i(h_{2,t}^{\mathcal{C}}|\tilde{\sigma}_t^i, \sigma_t^{-i}, d^*) \right] \Big| \mathcal{H}_t \right\}$$

$$\overset{③}{=} \mathbb{E}_{\mathcal{H}_t} \left\{ \mathbb{E}_{\tilde{d}_t, d^*} \mathbb{E}_{\mathcal{T}_{i,t}} \left[ \sum_{j=1}^{m_{i,t}} \sum_{a \in \alpha(h_{j,t}^{\mathcal{C}})} (\tilde{c}_t(h_{j,t}^{\mathcal{C}}, a) - c^*(h_{j,t}^{\mathcal{C}}, a)) u^i(h_{j,t}^{\mathcal{C}}|\tilde{\sigma}_t^i, \sigma_t^{-i}, \tilde{d}_t) \right] \Big| \mathcal{H}_t \right\}$$

$$+ \mathbb{E}_{\mathcal{H}_t} \left\{ \mathbb{E}_{\tilde{d}_t, d^*} \mathbb{E}_{\mathcal{T}_{i,t}} \left[ u^i(z_{i,t}|\tilde{r}_t) - u^i(z_{i,t}|r^*) \right] \Big| \mathcal{H}_t \right\}.$$

Here equation ① holds by inserting a term $c^*(h_{1,t}^{\mathcal{C}}, a)u^i(h_{1,t}^{\mathcal{C}}|\tilde{\sigma}_t^i, \sigma_t^{-i}, \tilde{d}_t)$; equation ② holds since that $h_{2,t}^{\mathcal{C}}$ is reached following the underlying transition $c^*(h_{1,t}^{\mathcal{C}}$; and equation ③ holds by using induction.

Therefore we finish the proof. $\qquad\square$

We upper bound the term $u^i(z_{i,t}|\tilde{r}_t) - u^i(z_{i,t}|r^*)$ first. We can refer to the technique of previous work in PSRL. Recall that in episode $t$, players reaches terminal node $z_{i,t}$ with a visited count $n_t(z_{i,t})$. We denote that $\bar{u}_t^i(z_{i,t})$ as the empirical mean of $u^i(z_{i,t}|r^*)$. Simply we insert $\bar{u}_t^i(z_{i,t})$ to get

$$u^i(z_{i,t}|\tilde{r}_t) - u^i(z_{i,t}|r^*) \le |u^i(z_{i,t}|\tilde{r}_t) - \bar{u}_t^i(z_{i,t})| + |\bar{u}_t^i(z_{i,t}) - u^i(z_{i,t}|r^*)|.$$

First we consider the second one $|\bar{u}_t^i(z_{i,t}) - u^i(z_{i,t}|r^*)|$ and we have similar bound for the first term. Conditioning on $r^*(z_{i,t})$, we can apply the Chernoff-Hoeffding bound (Hoeffding, 1994). For $\delta \in (0,1)$

$$Pr\left(|\bar{u}_t^i(z_{i,t}) - u^i(z_{i,t}|r^*)| \ge \sqrt{\frac{\ln(2/\delta)}{2\max(n_t(z_{i,t}),1)}}\Big|r^*(z_{i,t})\right) \le \delta, \tag{14}$$

where $Pr$ denote the probability.

Then we use the above inequality to get below lemma:

**Lemma 5.** *At episode $t$, the expectation of $|\bar{u}_t^i(z_{i,t}) - u^i(z_{i,t}|r^*)|$ can be bounded by*

$$\mathbb{E}_{\mathcal{H}_t}\left\{\mathbb{E}_{\tilde{d}_t,d^*}\mathbb{E}_{\mathcal{T}_{i,t}}\left[|\bar{u}_t^i(z_{i,t}) - u^i(z_{i,t}|r^*)|\right]\Big|\mathcal{H}_t\right\}$$

$$\le \mathbb{E}_{\mathcal{H}_t}\left\{\mathbb{E}_{\tilde{d}_t,d^*}\mathbb{E}_{\mathcal{T}_{i,t}}\left[\sqrt{\frac{2\log(2/\delta)}{\max(n_t(z_{i,t}),1)}}\right]\Big|\mathcal{H}_t\right\} + 2|Z|\delta.$$

*Proof.* Notice that Eq. 14 holds conditioning on $r^*(z_{i,t})$ and the expectation is taken over the prior $\mathbb{P}_0$. Then we need to carefully apply the Eq. 14. For the convenience of notation, we use $\pi_t(h|d^*)$ to represent $\pi_{\tilde{\sigma}_t^i, \sigma_t^{-i}}(h|d^*)$. We further use $\mathbb{I}(\cdot)$ to indicate the identical function. Then we expand the expectation into integration

$$\mathbb{E}_{\mathcal{H}_t}\left\{\mathbb{E}_{\tilde{d}_t,d^*}\mathbb{E}_{\mathcal{T}_{i,t}}\left[|\bar{u}_t^i(z_{i,t}) - u^i(z_{i,t}|r^*)|\right]\Big|\mathcal{H}_t\right\}$$

$$\overset{①}{=} \sum_{z\in Z}\int |\bar{u}_t^i(z) - u^i(z|r^*)|\pi_t(z|d^*)Pr(d^*, d_t|\mathcal{H}_t)Pr(\mathcal{H}_t)d(d^*, d_t, \mathcal{H}_t)$$

$$\overset{②}{\le} \sum_{z\in Z}\int \sqrt{\frac{\ln(2/\delta)}{2\max(n_t(z),1)}}\pi_t(z|d^*)Pr(d^*, d_t|\mathcal{H}_t)Pr(\mathcal{H}_t)d(d^*, d_t, \mathcal{H}_t)$$

$$+ \sum_{z\in Z}\int 2\mathbb{I}\left(|\bar{u}_t^i(z) - u^i(z|r^*)| \ge \sqrt{\frac{\ln(2/\delta)}{2\max(n_t(z),1)}}\right)Pr(d^*|\mathcal{H}_t)Pr(\mathcal{H}_t)d(d^*, \mathcal{H}_t) \tag{15}$$

$$\overset{③}{=} \mathbb{E}_{\mathcal{H}_t}\left\{\mathbb{E}_{\tilde{d}_t,d^*}\left[\sqrt{\frac{2\log(2/\delta)}{\max(n_t(z_{i,t}),1)}}\right]\Big|\mathcal{H}_t\right\}$$

$$+ \sum_{z\in Z}\int 2\mathbb{I}\left(|\bar{u}_t^i(z) - u^i(z|r^*)| \ge \sqrt{\frac{\ln(2/\delta)}{2\max(n_t(z),1)}}\right)Pr(\mathcal{H}_t|d^*)\mathbb{P}_0(d^*)d(d^*, \mathcal{H}_t)$$

$$\overset{④}{\le} \mathbb{E}_{\mathcal{H}_t}\left\{\mathbb{E}_{\tilde{d}_t,d^*}\left[\sqrt{\frac{2\log(2/\delta)}{\max(n_t(z_{i,t}),1)}}\right]\Big|\mathcal{H}_t\right\} + 2|Z|\delta.$$

Here we extend the expectation into integration in equation ①. Inequality ② holds by separating the integral space. Equation ③ uses the Bayes rule and thus we can apply Eq. (14) to get equation ④.

Therefore we finish the proof. $\qquad\square$

For another term $|u^i(z_{i,t}|\tilde{r}_t) - \bar{u}^i_t(z_{i,t})|$, we can still apply the technique in Lemma 5 to get below lemma:

**Lemma 6.** *At episode $t$, the expectation of $|u^i(z_{i,t}|\tilde{r}_t) - \bar{u}^i_t(z_{i,t})|$ can be bounded by*

$$\mathbb{E}_{\mathcal{H}_t}\left\{\mathbb{E}_{\tilde{d}_t,d^*}\mathbb{E}_{\mathcal{T}_{i,t}}\left[|u^i(z_{i,t}|\tilde{r}_t) - \bar{u}^i_t(z_{i,t})|\right]\Big|\mathcal{H}_t\right\} \leq \mathbb{E}_{\mathcal{H}_t}\left\{\mathbb{E}_{\tilde{d}_t,d^*}\left[\sqrt{\frac{2\log(2/\delta)}{\max(n_t(z_{i,t}),1)}}\right]\Big|\mathcal{H}_t\right\} + 2|Z|\delta.$$

*Proof.* We can directly prove that

$$\mathbb{E}_{\mathcal{H}_t}\left\{\mathbb{E}_{\tilde{d}_t,d^*}\mathbb{E}_{\mathcal{T}_{i,t}}\left[|u^i(z_{i,t}|\tilde{r}_t) - \bar{u}^i_t(z_{i,t})|\right]\Big|\mathcal{H}_t\right\}$$

$$= \sum_{s\in Z}\int |u^i(s|\tilde{r}_t) - \bar{u}^i_t(s)|\pi_t(s|d^*)Pr(d^*,d_t|\mathcal{H}_t)Pr(\mathcal{H}_t)d(d^*,d_t,\mathcal{H}_t)$$

$$\leq \sum_{s\in Z}\int \sqrt{\frac{\ln(2/\delta)}{2\max(n_t(s),1)}}\pi_t(s|d^*)Pr(d^*,d_t|\mathcal{H}_t)Pr(\mathcal{H}_t)d(d^*,d_t,\mathcal{H}_t)$$

$$+ \sum_{s\in Z}\int 2\mathbb{I}\left(|u^i(s|\tilde{r}_t) - \bar{u}^i_t(s)| \geq \sqrt{\frac{\ln(2/\delta)}{2\max(n_t(s),1)}}\right)Pr(\tilde{d}|\mathcal{H}_t)Pr(\mathcal{H}_t)d(d^*,\mathcal{H}_t) \quad (16)$$

$$= \mathbb{E}_{\mathcal{H}_t}\left\{\mathbb{E}_{\tilde{d}_t,d^*}\left[\sqrt{\frac{2\log(2/\delta)}{\max(n_t(z_{i,t}),1)}}\right]\Big|\mathcal{H}_t\right\}$$

$$+ \sum_{s\in Z}\int 2\mathbb{I}\left(|u^i(s|\tilde{r}_t) - \bar{u}^i_t(s)| \geq \sqrt{\frac{\ln(2/\delta)}{2\max(n_t(s),1)}}\right)Pr(\mathcal{H}_t|d^*)\mathbb{P}_0(d^*)d(d^*,\mathcal{H}_t)$$

$$\leq \mathbb{E}_{\mathcal{H}_t}\left\{\mathbb{E}_{\tilde{d}_t,d^*}\left[\sqrt{\frac{2\log(2/\delta)}{\max(n_t(z_{i,t}),1)}}\right]\Big|\mathcal{H}_t\right\} + 2|Z|\delta.$$

The above proof is almost the same as that in Lemma 5 expect inequality (16). Since $d^*$ and $\tilde{d}_t$ are identically distributed conditioning on $\mathcal{H}_t$, then we apply below equality to Eq. (16):

$$\mathbb{I}\left(|u^i(s|\tilde{r}_t) - \bar{u}^i_t(s)| \geq \sqrt{\frac{\ln(2/\delta)}{2\max(n_t(s),1)}}\right)Pr(\tilde{d}_t|\mathcal{H}_t)Pr(\mathcal{H}_t)$$

$$= \mathbb{I}\left(|u^i(s|r^*) - \bar{u}^i_t(s)| \geq \sqrt{\frac{\ln(2/\delta)}{2\max(n_t(s),1)}}\right)Pr(d^*|\mathcal{H}_t)Pr(\mathcal{H}_t).$$

Then we finish the proof. $\qquad\square$

Hence we combine the results in Lemma 5 and 6 and get the conclusion that for any $\delta \in (0,1)$,

$$\mathbb{E}_{\mathcal{H}_t}\left\{\mathbb{E}_{\tilde{d}_t,d^*}\mathbb{E}_{\mathcal{T}_{i,t}}\left[u^i(z_{i,t}|\tilde{r}_t) - u^i(z_{i,t}|r^*)\right]\Big|\mathcal{H}_t\right\} \leq \mathbb{E}_{\mathcal{H}_t}\left\{\mathbb{E}_{\tilde{d}_t,d^*}\left[2\sqrt{\frac{2\log(2/\delta)}{\max(n_t(z_{i,t}),1)}}\right]\Big|\mathcal{H}_t\right\} + 4|Z|\delta.$$

Using a pigeon-hole principle and choosing $\delta = 1/(|Z|T)$, we have below lemma:

**Lemma 7.** *At episode $T$, the expectation of the summation of $u^i(z_{i,t}|\tilde{r}_t) - u^i(z_{i,t}|r^*)$ over the prior distribution $\mathbb{P}_0$ has an order of:*

$$\mathbb{E}_{\mathbb{P}_0}\left[\sum_{t=1}^T u^i(z_{i,t}|\tilde{r}_t) - u^i(z_{i,t}|r^*)\right] = O(\sqrt{|Z|T\ln(|Z|T)}).$$

Then we consider chance player node $h_{j,t}^{\mathcal{C}}$. We also denote $\bar{c}(h_{j,t}^{\mathcal{C}}, a)$ as the empirical mean of chance player's probability to choose $a$ at $h_{j,t}^{\mathcal{C}}$. Notice that the utility is bounded in $[-1, 1]$. We have

$$\sum_{a \in \alpha(h_{j,t}^{\mathcal{C}})} (\tilde{c}_t(h_{j,t}^{\mathcal{C}}, a) - c^*(h_{j,t}^{\mathcal{C}}, a)) u^i(h_{j,t}^{\mathcal{C}} | \tilde{\sigma}_t^i, \sigma_t^{-i}, \tilde{d}_t)$$

$$\leq 2 \sum_{a \in \alpha(h_{j,t}^{\mathcal{C}})} |\tilde{c}_t(h_{j,t}^{\mathcal{C}}, a) - c^*(h_{j,t}^{\mathcal{C}}, a)|$$

$$\leq 2 \sum_{a \in \alpha(h_{j,t}^{\mathcal{C}})} |\tilde{c}_t(h_{j,t}^{\mathcal{C}}, a) - \bar{c}(h_{j,t}^{\mathcal{C}}, a)| + 2 \sum_{a \in \alpha(h_{j,t}^{\mathcal{C}})} |\bar{c}(h_{j,t}^{\mathcal{C}}, a) - c^*(h_{j,t}^{\mathcal{C}}, a)|.$$

Then conditioning on $c^*(h_{j,t}^{\mathcal{C}}, a)$, we use the concentration bound for $L_1$ norm (i.e. the deviation inequality (Weissman et al., 2003) to get that for $\delta \in (0, 1)$

$$Pr\left( \sum_{a \in \alpha(h_{j,t}^{\mathcal{C}})} |\bar{c}(h_{j,t}^{\mathcal{C}}, a) - c^*(h_{j,t}^{\mathcal{C}}, a)| \geq \sqrt{\frac{2 \ln(2^A/\delta)}{\max(n_t(h_{j,t}^{\mathcal{C}}), 1)}} \Big| c^*(h_{j,t}^{\mathcal{C}}, a) \right) < \delta.$$

Similar to the analysis in $r$, we give below lemma:

**Lemma 8.** *At episode $t$, the expectation of $|\bar{c}(h_{j,t}^{\mathcal{C}}, a) - c^*(h_{j,t}^{\mathcal{C}}, a)|$ can be bounded by*

$$\sum_{a \in \alpha(h_{j,t}^{\mathcal{C}})} \mathbb{E}_{\mathcal{H}_t} \left\{ \mathbb{E}_{\tilde{d}_t, d^*} \mathbb{E}_{\mathcal{T}_{i,t}} \left[ |\bar{c}(h_{j,t}^{\mathcal{C}}, a) - c^*(h_{j,t}^{\mathcal{C}}, a)| \right] \Big| \mathcal{H}_t \right\}$$

$$\leq \mathbb{E}_{\mathcal{H}_t} \left\{ \mathbb{E}_{\tilde{d}_t, d^*} \left[ \sqrt{\frac{2 \ln(2^A/\delta)}{\max(n_t(h_{j,t}^{\mathcal{C}}), 1)}} \right] \Big| \mathcal{H}_t \right\} + |H^{\mathcal{C}}|\delta.$$

*Proof.* We use similar techniques to get

$$\mathbb{E}_{\mathcal{H}_t} \left\{ \mathbb{E}_{\tilde{d}_t, d^*} \mathbb{E}_{\mathcal{T}_{i,t}} \left[ \sum_{a \in \alpha(h_{j,t}^{\mathcal{C}})} |\bar{c}(h_{j,t}^{\mathcal{C}}, a) - c^*(h_{j,t}^{\mathcal{C}}, a)| \right] \Big| \mathcal{H}_t \right\}$$

$$= \sum_{h \in H^{\mathcal{C}}} \int \sum_{a \in \alpha(h)} |\bar{c}(h, a) - c^*(h, a)| \pi_t(h|d^*) Pr(d^*, d_t|\mathcal{H}_t) Pr(\mathcal{H}_t) d(d^*, d_t, \mathcal{H}_t)$$

$$\leq \sum_{h \in H^{\mathcal{C}}} \int \sqrt{\frac{2 \ln(2^A/\delta)}{\max(n_t(h), 1)}} \pi_t(s|d^*) Pr(d^*, d_t|\mathcal{H}_t) Pr(\mathcal{H}_t) d(d^*, d_t, \mathcal{H}_t)$$

$$+ \sum_{h \in H^{\mathcal{C}}} \int \mathbb{I}\left( \sum_{a \in \alpha(h)} |\bar{c}(h, a) - c^*(h, a)| \geq \sqrt{\frac{2 \ln(2^A/\delta)}{\max(n_t(h), 1)}} \right) Pr(d^*|\mathcal{H}_t) Pr(\mathcal{H}_t) d(d^*, \mathcal{H}_t)$$

$$\tag{17}$$

$$= \mathbb{E}_{\mathcal{H}_t} \left\{ \mathbb{E}_{\tilde{d}_t, d^*} \left[ \sqrt{\frac{2 \ln(2^A/\delta)}{\max(n_t(h), 1)}} \right] \Big| \mathcal{H}_t \right\}$$

$$+ \sum_{h \in H^{\mathcal{C}}} \int \mathbb{I}\left( \sum_{a \in \alpha(h)} |\bar{c}(h, a) - c^*(h, a)| \geq \sqrt{\frac{2 \ln(2^A/\delta)}{\max(n_t(h), 1)}} \right) Pr(\mathcal{H}_t|d^*) \mathbb{P}_0(d^*) d(d^*, \mathcal{H}_t)$$

$$\leq \mathbb{E}_{\mathcal{H}_t} \left\{ \mathbb{E}_{\tilde{d}_t, d^*} \left[ \sqrt{\frac{2 \log(2^A/\delta)}{\max(n_t(z_{i,t}), 1)}} \right] \Big| \mathcal{H}_t \right\} + |H^{\mathcal{C}}|\delta.$$

The proof process above is similar to that of Lemma 5. $\qquad \square$

Once again, we get below lemma:

**Lemma 9.** *At episode $t$, the expectation of $|\tilde{c}(h_{j,t}^{\mathcal{C}}, a) - \bar{c}(h_{j,t}^{\mathcal{C}}, a)|$ can be bounded by*

$$\sum_{a \in \alpha(h_{j,t}^{\mathcal{C}})} \mathbb{E}_{\mathcal{H}_t} \left\{ \mathbb{E}_{\tilde{d}_t, d^*} \mathbb{E}_{\mathcal{T}_{i,t}} \left[ |\tilde{c}(h_{j,t}^{\mathcal{C}}, a) - \bar{c}(h_{j,t}^{\mathcal{C}}, a)| \right] \Big| \mathcal{H}_t \right\}$$

$$\leq \mathbb{E}_{\mathcal{H}_t} \left\{ \mathbb{E}_{\tilde{d}_t, d^*} \left[ \sqrt{\frac{2 \ln(2^A/\delta)}{\max(n_t(h_{j,t}^{\mathcal{C}}), 1)}} \right] \Big| \mathcal{H}_t \right\} + |H^{\mathcal{C}}|\delta.$$

*Proof.* We use similar techniques to get

$$\mathbb{E}_{\mathcal{H}_t} \left\{ \mathbb{E}_{\tilde{d}_t, d^*} \mathbb{E}_{\mathcal{T}_{i,t}} \left[ \sum_{a \in \alpha(h_{j,t}^{\mathcal{C}})} |\tilde{c}(h_{j,t}^{\mathcal{C}}, a) - \bar{c}(h_{j,t}^{\mathcal{C}}, a)| \right] \Big| \mathcal{H}_t \right\}$$

$$= \sum_{h \in H^{\mathcal{C}}} \int \sum_{a \in \alpha(h)} |\tilde{c}(h, a) - \bar{c}(h, a)| \pi_t(h|d^*) Pr(d^*, d_t|\mathcal{H}_t) Pr(\mathcal{H}_t) d(d^*, d_t, \mathcal{H}_t)$$

$$\leq \sum_{h \in H^{\mathcal{C}}} \int \sqrt{\frac{2 \ln(2^A/\delta)}{\max(n_t(h), 1)}} \pi_t(s|d^*) Pr(d^*, d_t|\mathcal{H}_t) Pr(\mathcal{H}_t) d(d^*, d_t, \mathcal{H}_t)$$

$$+ \sum_{h \in H^{\mathcal{C}}} \int \mathbb{I} \left( \sum_{a \in \alpha(h)} |\tilde{c}(h, a) - \bar{c}(h, a)| \geq \sqrt{\frac{2 \ln(2^A/\delta)}{\max(n_t(h), 1)}} \right) Pr(\tilde{d}|\mathcal{H}_t) Pr(\mathcal{H}_t) d(d^*, \mathcal{H}_t)$$

(18)

$$= \mathbb{E}_{\mathcal{H}_t} \left\{ \mathbb{E}_{\tilde{d}_t, d^*} \left[ \sqrt{\frac{2 \ln(2^A/\delta)}{\max(n_t(h), 1)}} \right] \Big| \mathcal{H}_t \right\}$$

$$+ \sum_{h \in H^{\mathcal{C}}} \int \mathbb{I} \left( \sum_{a \in \alpha(h)} |\bar{c}(h, a) - c^*(h, a)| \geq \sqrt{\frac{2 \ln(2^A/\delta)}{\max(n_t(h), 1)}} \right) Pr(\mathcal{H}_t|d^*) \mathbb{P}_0(d^*) d(d^*, \mathcal{H}_t)$$

$$\leq \mathbb{E}_{\mathcal{H}_t} \left\{ \mathbb{E}_{\tilde{d}_t, d^*} \left[ \sqrt{\frac{2 \log(2^A/\delta)}{\max(n_t(z_{i,t}), 1)}} \right] \Big| \mathcal{H}_t \right\} + |H^{\mathcal{C}}|\delta.$$

The proof process here is similar to that of Lemma 6. $\qquad\square$

Next, we use a pigeon-hole principle and choosing $\delta = 1/(|H^{\mathcal{C}}|T)$, we have below lemma:

**Lemma 10.** *At episode $T$, the expectation of the summation of $|\tilde{c}_t(h_{j,t}^{\mathcal{C}}, a) - c^*(h_{j,t}^{\mathcal{C}}, a)|$ over the prior distribution $\mathbb{P}_0$ has an order of:*

$$\mathbb{E}_{\mathbb{P}_0} \left[ \sum_{t=1}^{T} \sum_{j=1}^{m_{i,t}} \sum_{a \in \alpha(h)} |\tilde{c}_t(h_{j,t}^{\mathcal{C}}, a) - c^*(h_{j,t}^{\mathcal{C}}, a)| \right] = O(\sqrt{|H^{\mathcal{C}}|D^{\mathcal{C}} AT \ln(|H^{\mathcal{C}}|T)}).$$

Therefore we use the conclusion in Lemma7 and 10 to get

$$\sum_{t=1}^{T} \mathbb{E}_{\mathcal{H}_t} \left\{ \mathbb{E}_{\tilde{d}_t, d^*} \left[ u^i(\tilde{\sigma}_t^i, \sigma_t^{-i}|\tilde{d}_t) - u^i(\tilde{\sigma}_t^i, \sigma_t^{-i}|d^*) \right] \Big| \mathcal{H}_t \right\} = O(\sqrt{|Z|T \ln(|Z|T)} + \sqrt{|H^{\mathcal{C}}|D^{\mathcal{C}} AT \ln(|H^{\mathcal{C}}|T)}).$$

The similar proof can be applied to the second term to get the same upper bound by simply replacing $\tilde{d}_t$ with $d_t$:

$$\sum_{t=1}^{T} \mathbb{E}_{\mathcal{H}_t} \left\{ \mathbb{E}_{\tilde{d}_t, d^*, d_t} \left[ u^i(\tilde{\sigma}_t^i, \sigma_t^{-i}|d^*) - u^i(\tilde{\sigma}_t^i, \sigma_t^{-i}|d_t) \right] \Big| \mathcal{H}_t \right\} = O(\sqrt{|Z|T \ln(|Z|T)} + \sqrt{|H^{\mathcal{C}}|D^{\mathcal{C}} AT \ln(|H^{\mathcal{C}}|T)}).$$

Sum the analysis together, we get to the conclusion that

$$\mathbb{E}_{H_T} \left\{ \mathbb{E}_{d^*, d_T} \left[ \mathcal{G}_T^i \Big| H_T \right] \right\} = O(\sqrt{\frac{|Z| \ln(|Z|T)}{T}} + \sqrt{\frac{|H^{\mathcal{C}}|D^{\mathcal{C}} A \ln(|H^{\mathcal{C}}|T)}{T}})$$

