# OpenReview forum: "Posterior sampling for multi-agent reinforcement learning: solving extensive games with imperfect information"
_ICLR.cc/2020/Conference — Accept (Talk)_

### Official Review · AnonReviewer3 · 2019-10-17
**Official Blind Review #3**

**Rating:** 8

**Review:**

Posterior sampling for multi-agent reinforcement learning: solving extensive games with imperfect information
================================================================


This paper investigates the use of Thompson sampling in multi-agent reinforcement learning.
They present a natural extension of the PSRL algorithm paired with counterfactual regret minimization, rather than expected reward maximization.
They provide support for this algorithm's efficacy through a theorem that proves polynomial learning rates, together with empirical evaluation where this approach is competitive with state of the art.


There are several things to like about this paper:
- This paper is definitely "groundbreaking" in that it makes a true extension to the existing literature: PSRL has been relatively well-studied in single-agent RL but never (to my knowledge) in the multi-agent setting.
- The extensions from single agent to multi-agent are natural, but also non-trivial, and it seems like this is a genuinely novel piece of work that can be interesting to both side (exploration and multi-agent).
- The general structure of the paper and presentation is good.
- The support from the theorem is great, and also the empirical evaluation is convincing.

There are a few places where the paper might be improved:
- It might be helpful to draw the connection to Thompson sampling more explicitly at the start. PSRL is really an application of Thompson sampling principle, but it is important that it doesn't happen every step but instead on a longer timescale. It might be helpful to cite "a tutorial on thompson sampling" Russo et al.
- Do you think there are promising avenues towards PSRL with generalization (rather than tabular)? It feels like actually this should carry over naturally... so maybe you should mention this?
- I'm not really an expert on the novelty / impressiveness of this algorithm in the multi-agent setting so cannot fully comment on that.


Overall I think this is a really interesting paper that should be of interest to the ICLR community.
I can't say this with full confidence (especially with respect to the multi-agent side) but I do think it's something that probably would add value to the conference!


**Experience Assessment:**

I have published one or two papers in this area.

**Review Assessment: Checking Correctness Of Derivations And Theory:**

I assessed the sensibility of the derivations and theory.

**Review Assessment: Checking Correctness Of Experiments:**

I assessed the sensibility of the experiments.

**Review Assessment: Thoroughness In Paper Reading:**

I read the paper at least twice and used my best judgement in assessing the paper.

---

> ### Author Response · Authors · 2019-11-12
> **Response to Reviewer 3**
>
>
> To Reviewer 3:
> Thank you for acknowledging our novel contributions as well as providing valuable suggestions.
>
> Q1: Connection with Thompson Sampling (TS):
> Thanks for the suggestion. We have revised the introduction to include it.
>
> Q2: On PSRL with generalization:
> Thanks. PSRL with generalization is a very interesting problem but it is also very challenging. It is worth of a systematic investigation to bridge the gap between the provable tabular RL algorithms and PSRL methods with generalization. Bootstrapping might be one possible direction. [1] applies the principle of PSRL to DQN by using bootstrapping. Another possible direction is to adapt more practical Bayesian inference algorithms to RL tasks.
>
> [1] Osband, Ian, et al. "Deep exploration via bootstrapped DQN." Advances in neural information processing systems. 2016.

---

> > ### Comment · AnonReviewer3 · 2019-11-14
> > **Sounds good**
> >
> > I think that it makes sense not to address that generalization in this paper, but I still think it might be worth adding a mention of this issue (+ maybe even suggested avenues as [1]) in your paper.
> >
> > I think this could end up prompting lots of interesting follow-on work in the field... and so will help to increase the impact of this paper to call some of these out.

---

> > > ### Author Response · Authors · 2019-11-15
> > > **Response to Reviewer 3**
> > >
> > > Thanks for your nice suggestion. We have added the discussion on PSRL with generalization in Sec. 6.

---

### Official Review · AnonReviewer2 · 2019-10-26
**Official Blind Review #2**

**Rating:** 6

**Review:**

PSRL
------

This work considers the task of finding a Nash equilibrium in a two-player zero-sum imperfect information game, where some aspects of the game are not known to the agents (specifically, the chance node probabilities, and the reward function).

The authors propose a method based on PSRL, i.e. at each iteration a set of game parameters are sampled from the posterior of the distribution. Then, CFR is applied in the inner loop; but instead of finding the NE strategy, one player finds the strategy that basically maximizes the reward deviation between two games sampled from the posterior, given that the opponent is playing Nash in the first game.

The authors prove convergence bounds for their algorithm, and demonstrate its performance on Leduc Hold'em with game parameters randomly chosen from a Dirichlet distribution.

-------------------------

I agree with the authors that standard CFR suffers from the requirement that the full game is known, so it doesn't work well in its standard form when the environment is not known. There *are* other regret minimizers that do work in the model-free setting ([1], [2], [3]), none of which are discussed by the authors.

I also find the proposed setting somewhat unconvincing. The authors are considering a situation where the environment is unknown, but it's not the RL setting because you still must control both (opposing!) agents. Of course, you can't actually find a Nash Equilibrium if you can't control the other player (because they might just never explore part of the game tree). But you would want your algorithm to be a regret minimizer regardless of your partner's strategy. Is this true of the proposed algorithm?

The proposed interaction strategy described in Eq. 5 and 6 is clever: basically, each player explores a part of the tree that maximizes the difference in payoffs between the two sampled games. This seems a bit inefficient thouggh, why doesn't the agent just find the BR to \sigma_{-i} under \tilde{d} given that the opponent plays the NE under d? I don't see why you would want to explore parameters that have a high reward uncertainty if there's a different strategy that does better than the whole confidence interval. It's like UCB: you should play a strategy not with the highest uncertainty, but with the highest optimistic payoff.

I think the work could be substantially improved by comparing against model-free baselines, e.g. fictitious self-play [2], CFR with outcome sampling [1]. The current work deosn't provide any evidence of what benefits the Bayesian approach provides over model-free regret minimization. Especially since the Bayesian approach presumably does not scale as well due to the requirement of maintaining beliefs over all possible games, and the requirement that a correct prior is provided. I would be curious to see Bayesian and model-free appraoches compared in games of different sizes to see how the different methods scale.

Nits:
- such as the private pokers in poker games (private cards)
- h_1, h_1 \in H^C, are independent
- The reference to Neil 2018 should be "Neil Burch", not "Burch Neil"
- And if you're going to say "we can directly apply the technique from [100-page PhD thesis]", please mention the page number.



[1] Lanctot, Marc, et al. "Monte Carlo sampling for regret minimization in extensive games." Advances in neural information processing systems. 2009.
[2] Heinrich, Johannes, Marc Lanctot, and David Silver. "Fictitious self-play in extensive-form games." International Conference on Machine Learning. 2015.
[3] Srinivasan, Sriram, et al. "Actor-critic policy optimization in partially observable multiagent environments." Advances in Neural Information Processing Systems. 2018.




**Experience Assessment:**

I have published one or two papers in this area.

**Review Assessment: Checking Correctness Of Derivations And Theory:**

I assessed the sensibility of the derivations and theory.

**Review Assessment: Checking Correctness Of Experiments:**

I carefully checked the experiments.

**Review Assessment: Thoroughness In Paper Reading:**

I read the paper thoroughly.

---

> ### Author Response · Authors · 2019-11-12
> **Response to Reviewer 2 (part 1)**
>
> Thank you very much for your valuable questions and suggestions. Below, we address your concerns starting with a potential misunderstanding.
>
> Q1: A potential misunderstanding:
> You commented that “given that the opponent is playing Nash in the first game”. But this is not a correct description of our algorithm. Actually, the opponent is playing the strategy generated by CFR in the last round, i.e., \sigma^{-i}_t, instead of the average strategy (the approximate Nash). We have polished our description on the interaction strategy in Sec.3 to make it clearer.
>
> Q2: On the regret minimizers in the model-free setting and missing baselines:
> Thanks for the kind suggestion. We agree that it is very valuable to compare our method with Fictitious self-play (FSP) [3] and MCCFR with outcome sampling (MCCFR-OS) [2]. We have included them and added comparison in the revision.
>
> In fact, we have already discussed the FSP in Sec. 4 and evaluated our algorithm against FSP-PSRL in the first submission, and our algorithm shows a better convergence rate (See Sec. 5, Fig. 2). We have added FSP-Fitted-Q using the same hyperparameters as reported in [3], as a new baseline method in Sec.5. And FSP-Fitted-Q converges much slower than FSP-PSRL. Also, we have added the discussion on MCCFR-OS in Sec. 4 and done additional experiments on MCCFR-OS in Sec. 5 of the revision. In our implementation of MCCFR-OS, we use epsilon-greedy to do exploration as suggested in [2]. We set epsilon=0.1. The performances of FSP-Fitted-Q and MCCFR-OS are very poor compared with other algorithms. This may be because (1) a model-free algorithm is very unusual to outperform model-based methods in terms of sample complexity on tabular RL tasks (Please refer to [4], Sec.4.5 of [5] for some related results); (2) they don’t have efficient exploration strategies: but to the best of our knowledge, there exists few principled ways to do exploration efficiently in MARL even on tabular tasks. And developing a method, which is efficient at least in theory, to do exploration is exactly the motivation of our work.
>
> As for another related work [6] you mentioned, we have included it in the related work. We didn’t include it as a baseline method since we concentrate on the tabular-based methods to compare their exploration efficiency. The use of functional approximators in [6] involves extra learning error and the comparison is less significant.
>
> Q3: About the proposed setting:
> Thanks. Indeed, controlling both agents is not reasonable if you are using the trained strategy to play with an opponent which you cannot control. But controlling both agents in training is very standard and widely used. Previous work on Markov games under the RL setting [1] (which is a general setting in MARL) also assumes both agents to be controlled by one algorithm in the training phase to make sure that the generated strategy approaches the Nash Equilibrium. This setting is also used in the fictitious self-play and MCCFR with outcome sampling that you mentioned. In Sec.2 of the revision, we have refined our description for the setting.

---

> > ### Author Response · Authors · 2019-11-12
> > **Response to Reviewer 2 (part 2)**
> >
> >
> > Q4: On our interaction strategy and the high optimistic payoff interaction strategy:
> > Thanks for the kind suggestion, but it doesn’t work as expected.
> >
> > First, we’d like to provide more insights on how we design the interaction strategy: Intuitively, \sigma_t is generated by CFR with a biased knowledge on d^*. So the target of the interaction strategy is to *fix the bias*. More formally, the “bias” can be described by the term \mathcal{G}_T^i, and we proved that our interaction strategy can make sure \mathcal{G}_T^i decrease in a speed of O(\sqrt{\log T/ T}). That’s why we design the interaction strategy in this way.
> >
> > Then, as for your comment “you should play a strategy not with the highest uncertainty, but with the highest optimistic payoff”, what you said is true in the setting of single agent reinforcement learning just like UCB as you commented. However, playing strategies with a high optimistic payoff may not be efficient in the setting of multi-agent reinforcement learning: if \sigma^{-i} is not a good strategy for the opponent, then playing (BR(\sigma^{-i}), \sigma^{-i}) may provide little information about the Nash where BR(\sigma^{-i}) is the best response of \sigma^{-i} under \tilde{d}_t, though player i can receive more payoffs in the simulation. And in our algorithm, \sigma_t^{-i}_t is not Nash. Instead, it is generated by CFR with a biased knowledge. Thus, \sigma_t^{-i} may not be a good strategy.
> >
> > According to the reason above, we failed to establish any theoretical connection between (BR(\sigma^{-i}_t), \sigma^{-i}_t) and the exploitability. Moreover, we have empirically evaluated the performance of (BR(\sigma^{-i}_t), \sigma^{-i}_t)  (Please see Fig.2 in the revision). And it does much worse than our algorithm.
> >
> >
> > References:
> > [1] Littman, Michael L. "Markov games as a framework for multi-agent reinforcement learning." Machine learning proceedings 1994. Morgan Kaufmann, 1994. 157-163.
> > [2] Lanctot, Marc, et al. "Monte Carlo sampling for regret minimization in extensive games." Neurips. 2009.
> > [3] Heinrich, Johannes, Marc Lanctot, and David Silver. "Fictitious self-play in extensive-form games." International Conference on Machine Learning. 2015.
> > [4] Sun, Wen, et al. "Model-based RL in contextual decision processes: PAC bounds and exponential improvements over model-free approaches." Conference on Learning Theory. 2019.
> > [5] Wiering, Marco, and Martijn Van Otterlo. "Reinforcement learning." Adaptation, learning, and optimization 12 (2012): 3.
> > [6] Srinivasan, Sriram, et al. "Actor-critic policy optimization in partially observable multiagent environments." Advances in Neural Information Processing Systems. 2018.

---

> > > ### Comment · AnonReviewer2 · 2019-11-14
> > > **Response**
> > >
> > > Thank you to the authors for their detailed reply, and for updating the paper. The response to Q4 is especially enlightening and interesting, and it would be helpful to include some of this intuition in the manuscript. I have updated my scores accordingly.
> > >
> > > Re Q2, it is true that much prior MARL work considers the "centralized training" setting, but isn't that typically because you assume that you can run training in a simulated (known) environment? I'm not sure in what TZIEGs you can control all the agents but don't have access to the environment... maybe 1-on-1 robot basketball in the real world? :)

---

> > > > ### Author Response · Authors · 2019-11-15
> > > > **Response to Reviewer 2**
> > > >
> > > > Thanks. We have included the intuition on our interaction strategy in the revision.
> > > >
> > > > About the setting: In fact, the robot basketball is indeed an example of our setting. Actually, the centralized training has been used in robot soccer [1]. In robot soccer, you can run training in a simulated environment, but you still cannot directly access to the parameters of the environment as the randomness on the transition and reward functions comes from the noisy sensors and actuators.
> > > >
> > > > [1] Peter Stone and Richard S. Sutton. Scaling reinforcement learning toward RoboCup soccer. ICML, 2001.

---

### Official Review · AnonReviewer4 · 2019-11-04
**Official Blind Review #4**

**Rating:** 6

**Review:**

Review for "Posterior Sampling for Multi-Agent Reinforcement Learning".

The paper proposes a sample-efficient way to compute a Nash equilibrium of an extensive form game. The algorithm works by maintaining a probability distribution over the chance player / reward pair (i.e. an environment model).

I give a weak recommendation to accept the paper. Although I haven't checked the proofs in detail, the premise seems to be sound - the authors extend model-based exploration results from MDPs to games. The essence of the argument seems to be that the model of the chance player becomes close to d^\star quickly enough to get a sub-linear bound.

The main complaints I have about the paper concern clarity.

1. The paper is very densely written. This isn't necessarily bad, but it makes the paper a bit hard to understand. It would benefit the manuscript greatly to provide a figure which shows how the algorithm works for a small toy game. There is space left in the paper, so even a one-page figure would fit in. The figure should show all the major quantities: d, \sigma, u.

2. The meaning of the quantity \mathcal{G}_T^i should be more thoroughly described, given it is important in the proof.

3. You define a game with N players, but the algorithm works with 2.

4. Do you really need all the notations in section 2.1? Why not just define the ones used in the algorithm?

5. Can you discuss how large the constants \xi can become in practice? The definition of \xi^i seems to be different on page 10 and in Theorem 1 - please disambiguate.

I ask the authors to add a figure and address the issues above.

I am not an expert in this sub-field so I may have missed aspects of the paper.

Minor points:
- In Figure 1, please say that "default" is your algorithm.
- "optimal in the face of uncertainty" => "optimism in the face of uncertainty"

**Experience Assessment:**

I have read many papers in this area.

**Review Assessment: Checking Correctness Of Derivations And Theory:**

I assessed the sensibility of the derivations and theory.

**Review Assessment: Checking Correctness Of Experiments:**

I assessed the sensibility of the experiments.

**Review Assessment: Thoroughness In Paper Reading:**

I read the paper at least twice and used my best judgement in assessing the paper.

---

> ### Author Response · Authors · 2019-11-12
> **Response to Reviewer 4:**
>
> Thanks for acknowledging our novel contributions as well as giving the valuable comments. We have revised the paper (especially sections 2&3) to give a clearer description of our work and removed unnecessary notations. Please refer to the revision. Below, we address the concerns in detail.
>
> Q1: Densely written and a figure for toy game:
> Thanks. We have revised the paper to make it clearer. We also added a figure for a toy game in the newly added Fig.1 (See Sec.3 of the revised paper).
>
> Q2: Quantity \mathcal{G}_T^i:
> Thanks. We have made it clearer in revision (See Section 3 after Eq. 4). Here, \mathcal{G}_T^i denotes the gap between exploitability and the regret from CFR. Minimizing \mathcal{G}_T^i leads to the reduction of uncertainty of the environment.
>
> Q3: N-player:
> Thanks. We have revised Section 2 to make it clear that our focus is on two-player zeros-sum games;
> we start with the extensive game with N players so as to provide a general definition and introduce related notations, which subsumes TZIEG as nontrivial special cases.
>
> Q4: Unnecessary notations:
> Thanks. We have checked the notations carefully and removed unnecessary ones.
>
> Q5: Constants \xi:
> Constants \xi are parameters related to the structure of the game. Our definition of \xi is from Corrollary 2 in [1]. Generally, we have the lower and upper bounds for \xi as: sqrt{|\mathcal{I}|} \leq \xi \leq |\mathcal{I}|. In most games, \xi is much smaller than |\mathcal{I}|, e.g., in No limit Texas Hold, \xi is about 10000 as shown in [1] while the size of the game tree is more than 10^13. We added this discussion in Section 3. We have checked Theorem1 and the appendix to ensure that they have the same definition for \xi^i.
>
> References:
> [1] Neil, Burch. "Time and space: Why imperfect information games are hard." (2018).

---

### Decision · Program_Chairs · 2019-12-19

**Decision:**

Accept (Talk)

**Comment:**

The paper extends posterior sampling to the multi-agent RL setting, and develops a novel algorithm with convergence guarantees to a Nash Equilibrium strategy in two-player zero sum games. Reviewers raised several questions, many of which were well addressed by the authors and which helped further clarify the approach and contribution of the paper. The paper is timely in that novel connections between Game Theory and RL are being explored in fruitful ways, and the paper provides valuable new insights and directions for future research.